# Implementation of aerosol-cloud interactions in the regional atmosphere-aerosol model COSMO-MUSCAT(5.0) and evaluation using satellite data

Sudhakar Dipu[1], Johannes Quaas[1], Ralf Wolke[2], Jens Stoll[2], Andreas Mühlbauer[3], Odran Sourdeval[1], Marc Salzmann[1], Bernd Heinold[2], and Ina Tegen[2]

[1]Institute for Meteorology, Universität Leipzig, Germany
[2]Leibniz Institute for Tropospheric Research, Germany
[3]FM Global Research, Norwood, MA, U.S.A

*Correspondence to:* Dipu Sudhakar (dipu.sudhakar@uni-leipzig.de)

**Abstract.** The regional atmospheric model Consortium for Small Scale Modeling (COSMO) coupled to the MultiScale Chemistry Aerosol Transport model (MUSCAT) is extended in this work to represent aerosol-cloud interactions. Previously, only one-way interactions (scavenging of aerosol and in-cloud chemistry) and aerosol-radiation interactions were included in this model. The new version allows for a microphysical aerosol effect on clouds. For this, we use the optional two-moment cloud microphysical scheme in COSMO and the online-computed aerosol information for cloud condensation nuclei (CCN) concentrations, replacing the constant CCN concentration profile. In the radiation scheme, we have implemented a droplet-size-dependent cloud optical depth, allowing now for aerosol-cloud-radiation interactions. To evaluate the models with satellite data, the Cloud Feedback Model Inter-comparison Project Observational Simulator Package (COSP) has been implemented. A case study has been carried out to understand the effects of the modifications, where the modified modeling system is applied over the European domain with a horizontal resolution of $0.25° \times 0.25°$. To reduce the complexity in aerosol cloud interactions only warm-phase clouds are considered. We found that the online coupled aerosol introduces significant changes for some cloud microphysical properties. The cloud effective radius shows an increase of 9.5%, and the cloud droplet number concentration is reduced by 21.5%.

## 1 Introduction

The quantification of aerosol cloud interactions in models continues to be a challenge (IPCC, 2013). Estimates of effective radiative forcing and assessments of the radiative effects due to aerosol cloud interactions to a large extent rely on numerical modeling. A large effort has been made to represent such effects in general circulation models (GCMs) (Penner et al., 2006; Quaas et al., 2009; Ghan et al., 2016). However, GCMs do not resolve the processes relevant for cloud dynamics well. Improving the understanding of processes of aerosol-cloud interactions thus largely relies on simulations with cloud-resolving models and large-eddy simulations (LES) (Ackerman et al., 2000, 2004; Xue and Feingold, 2006; Sandu et al., 2008; Seifert et al., 2015; Berner et al., 2013). However, LES often focus on case studies and use idealized boundary conditions and also

an idealized representation of the aerosol. This leads to uncertainties specifically because, when analyzing cloud systems, or cloud regimes, rather than individual clouds, aerosol-cloud-precipitation interaction processes often are buffered (**?**). Regional climate modeling is a powerful tool to overcome these limitations of small-domain idealized LES, and much higher resolutions are possible than for GCMs. Compared to LES that only simulates individual cloud systems, regional climate models are able to simulate the feedbacks between clouds and aspects of the large-scale circulation and its variability reasonably well. Even though regional models do not describe part of the large-scale feedbacks, may be considered a good optimal compromise (Bangert et al., 2011; Van den Heever and Cotton, 2007; Chapman et al., 2009; Forkel et al., 2015; Yang et al., 2012). A still often applied cloud microphysics parameterization in numerical weather prediction is a bulk, one-moment scheme (Kessler, 1969; Lin et al., 1983), which uses the specific mases for different hydrometeor species as prognostic variables. However, it cannot treat aerosol cloud interactions because it calculates only one moment of the size distribution, and does not carry information about size or number concentration of cloud droplets. In contrast, bin microphysical schemes numerically resolve the size spectrum and are thus able to predict the spatiotemporal behavior of a number of size categories for each hydrometeor type explicitly (Khain et al., 2000; Simmel et al., 2015). However, this approach is numerically very expensive especially when applied for regional atmospheric models. As a compromise between these two approaches, two-moment microphysical schemes can predict the number concentration of the liquid and ice hydrometeors, in addition to mass variables (Cotton et al., 1986; Meyers et al., 1997; Seifert and Beheng, 2006). Furthermore, numerous studies have shown that using two-moment schemes is a promising avenue in future operational forecast models (Reisner et al., 1998; Tao et al., 2003; Seifert and Beheng, 2006) and is also computationally efficient.

At present, several weather prediction and global models apply with two-moment cloud microphysical schemes. For example, the Weather Research and Forecasting (WRF) model is available with different types of two-moment microphysical schemes (Thompson et al., 2008; Morrison and Gettelman, 2008; Lim and Hong, 2010). Morrison et al. (2009) showed that using a two-moment scheme in the WRF model produced more trailing stratiform precipitation in an idealized two-dimensional squall case which is more consistent with observations. In another study, Li et al. (2008) investigated the effect of aerosol on cloud microphysical processes with a two-moment microphysical scheme in WRF model. Also, Lim and Hong (2010) have included a prognostic equation for cloud water and cloud condensation nuclei number concentration ($C_{ccn}$), which could reduce the uncertainty in investigating the aerosol effect on cloud properties and the precipitation process in WRF model. Seifert et al. (2012) and Weverberg et al. (2014) compared the operational one-moment and two-moment microphysical schemes in the Consortium for Small Scale Modeling atmospheric model (COSMO). Further, some groups previously implemented aerosol-cloud interactions in COSMO, albeit with a different aerosol scheme (Bangert et al., 2011; Zubler et al., 2011; Possner et al., 2015) and very few are coupled to the radiation scheme. Seifert et al. (2012) compared the operational one-moment microphysics scheme to a two-moment scheme. They found that aerosol perturbation have significant effect on radiation and near surface temperature, rather than the resulting surface precipitation.

In this paper, we discuss the improved cloud microphysics parameterization in the COSMO model (Doms and Schättler, 1999), via the online-coupled aerosol model, MUlti-Scale Chemistry-Aerosol Transport (MUSCAT, Wolke et al., 2004, 2012). The two-moment cloud microphysical scheme in the COSMO model (Seifert and Beheng, 2006) uses fixed profiles of $C_{ccn}$.

Rather than this simplification, here we use $C_{ccn}$ predicted on the basis of the simulated aerosol from the MUSCAT module. This will enable the COSMO model to have temporally and spatially varying $C_{ccn}$ at each grid point, which are fully consistent with the cloud and precipitation fields, as well as with dynamics (e.g. scavenging is taken into account, as is vertical transport) to represent aerosol cloud interactions. In two further steps, (i) the radiation scheme is slightly revised to consider the cloud droplet size information (so far considered constant even when applying the two-moment cloud microphysical scheme), and (ii) a diagnostic tool, the Cloud Feedback Model Intercomparison Project Observational Simulator Package (Bodas-Salcedo et al., 2011, 2008; Nam and Quaas, 2012) is implemented that allows for a consistent evaluation using satellite observations. The paper is organized as follows: section 2 gives a brief introduction to the coupled model system COSMO-MUSCAT, data, and methodology. The comparison between the improved two-moment cloud microphysical parameterization with the available two-moment scheme making use of the COSP satellite simulator is discussed in section 3. Finally, concluding remarks are found in section 4.

## 2  Data and Methodology

### 2.1  The COSMO-MUSCAT model and revised cloud activation

The non-hydrostatic three-dimensional model, COSMO, which was originally developed for limited-area operational numerical weather predictions (NWP), is used in this study (Doms and Schättler, 1999; Steppeler et al., 2003). This model has been used operationally in convection permitting configurations since 2007 by the German Weather Service (Deutscher Wetterdienst, DWD,  Baldauf et al., 2011). In this study, we have used COSMO version 5.0, which is initialized and forced by reanalyzed data provided by the global meteorological model GME (Global Model of the Earth) of DWD, which is a hydrostatic weather prediction model (Majewski et al., 2002). GME operates on an icosahedral-hexagonal grid having a horizontal resolution of approximately 40 km and vertical resolution of 40 layers up to 10 hPa. The COSMO model is initialized with the interpolated GME initial state and nested within GME with hourly updates of lateral boundary values. In this study, the COSMO model has been configured in a non-convection permitting mode with a uniform horizontal grid resolution of $0.25°$ ($\approx$28 km). The two-moment cloud microphysics scheme in the COSMO model (Seifert and Beheng, 2006) distinguishes between five hydrometeors classes, namely cloud droplets, rain, ice crystals, snow, and graupel. Processes in the warm (liquid) phase considered by this scheme include the nucleation of cloud droplets, autoconversion of cloud droplets to form rain, accretion, and self-collection of rain droplets. The formulations have been derived by Seifert and Beheng (2001) from the theoretical formulation of Beheng and Doms (1986). However, the radiation scheme does not yet make use of the additional information about cloud particle sizes provided by the two-moment microphysics. It uses the Ritter and Geleyn (1992) parameterization for the cloud optical properties in radiation scheme. According to Ritter and Geleyn (1992), the cloud optical properties are approximated by the relation between specific liquid water content $q_c$, and cloud effective radius $r_e$. Thus cloud optical depth $\tau_c$ is expressed as,

$$\tau_c = (c_1 + \frac{c_2}{r_e})q_c dz \tag{1}$$

where dz is layer thickness, and $c_1$ and $c_2$ are constants. Similarly, the effective radius $r_e$ is related to specific cloud water content and is approximated as,

$$r_e = c_3 + c_4 q_c \tag{2}$$

where $c_3$ and $c_4$ are constants (Ritter and Geleyn, 1992). In order to take in to account the two-moment microphysics in radiation scheme the cloud optical properties have to be modified. The cloud effective radius $r_e$ is derived by dividing the third and second moment of the size distribution (Martin et al., 1994; Morrison and Gettelman, 2008) which, after rearranging, yields,

$$r_e = \frac{\Gamma(\mu+4)}{2\lambda\Gamma(\mu+3)} \tag{3}$$

where $\mu$ is the spectral shape parameter (here, $\mu = 2$), $\Gamma$ is the gamma distribution function and $\lambda$ is the slope parameter, which is given by

$$\lambda = \left[\frac{\pi\rho_w N_d \Gamma(\mu+4)}{6\rho q_c \Gamma(\mu+1)}\right]^{\frac{1}{3}} \tag{4}$$

where $\rho$ is the density of the air, $\rho_w = 1000\,\text{kg}\,\text{m}^{-3}$ is the bulk density of liquid water, $N_d$ is the droplet number concentration, and $q_c$ is the specific water content. The corresponding cloud optical depth is given by

$$\tau_c = \frac{3\rho q_c dz}{2\rho_w r_e} \tag{5}$$

where, dz is the layer thickness, $\rho_w$ is the bulk density of liquid water and the droplet size spectrum is considered vertically constant in the grid layer.

The online coupled model system COSMO-MUSCAT (Wolke et al., 2004; Renner and Wolke, 2010; Wolke et al., 2012) is used for prognostic cloud condensation nuclei in the cloud microphysics parameterization in COSMO model. The chemistry/aerosol transport model MUSCAT treats atmospheric transport as well as chemical reactions, with the Regional Atmospheric Chemistry Mechanism (RACM, Stockwell et al., 1997). In MUSCAT, all meteorological fields are given with respect to the uniform horizontal meteorological grid from the online coupled COSMO2M (COSMO with two-moment scheme) model, whereas the aerosol information is fed back to the COSMO2M model from MUSCAT. In the previous setting, the interactions only considered the radiative effects of aerosols (scattering and absorption of solar radiation), as well as the scavenging of aerosol and in-cloud aerosol chemistry. A diagram illustrating the COSMO-MUSCAT modeling set up is shown in Figure 1. In the COSMO model with two-moment approach, the nucleation of cloud droplets has been treated explicitly and the aerosol activation parameterization is based on empirical activation spectra, which is in the form of power law,

$$N_{ccn} = C_{ccn} S^k, S\,in\,\% \tag{6}$$

where S is supersaturation, $C_{ccn} = 1.26 \times 10^9\,m^{-3}$, and k = 0.308 for continental conditions or $C_{ccn} = 1.0 \times 10^8\,m^{-3}$ and k = 0.462 for maritime conditions (Khain et al., 2001). Seifert et al. (2012) investigated the influence of substantially perturbing

$C_{ccn}$ from 100 to 3200 $\mathrm{cm}^{-3}$ (see above for a brief discussion of this paper). Accordingly, the grid scale explicit nucleation rate is calculated from the time derivative of the activation relation (Seifert and Beheng, 2006),

$$
\left.\frac{\partial N_c}{\partial t}\right|_{nuc} = \begin{cases} C_{ccn} k S^{k-1} \frac{\partial S}{\partial z} w, & \text{if } S \geq 0, w\frac{\partial S}{\partial z} > 0, \\ & \text{and } S < S_{max}, \\ 0 & \text{else} \end{cases} \tag{7}
$$

The above parameterization scheme uses constant $C_{ccn}$, and $S_{max}$ varies with atmospheric conditions (in maritime conditions, $C_{ccn}$ assumes that at $S_{max} = 1.1\%$, all $C_{ccn}$ are already activated). In the above equation, nucleation explicitly depends on grid scale supersaturation in combination with saturation adjustment assumed in the cloud scheme, which has limitations (Seifert and Beheng, 2006). As an initial step, a coupled model simulation is carried out by setting $S_{max} = 2.0\%$, the condition for intermediate aerosols in COSMO model. In a second step, we have used simulated sulfate ($SO_4$) aerosol mass concentration information from the MUSCAT model. The emission inventory in the MUSCAT model is provided by the TNO (European emissions processing, and stands for Nederlandse Organisatie voor toegepast-natuurwetenschappelijk onderzoek (Netherlands Organisation for Applied Scientific Research)) for the Air Quality Model Evaluation International Initiative (AQMEII) project (Pouliot et al., 2012). $C_{ccn}$ is derived using the following empirical relation (Boucher and Lohmann, 1995),

$$
C_{ccn} = 10^{2.21 + 0.41 \log(mSO_4)} \tag{8}
$$

where $mSO_4$ is the sulfate aerosol mass concentration in $\mu\mathrm{g\,m}^{-3}$. The constant $C_{ccn}$ in equation (7) is replaced by the spatially and temporally varying $C_{ccn}$ values, derived from equation (8), using the sulfate aerosol mass concentration from the MUSCAT module. This empirical relationship which links sulfate aerosol mass concentration to $C_{ccn}$ is subject to substantial uncertainty. Representing sulfate aerosol as a surrogate for all aerosols is probably too simplistic to capture the complexity of the whole activation process. Future work will introduce a more complex aerosol-cloud coupling, taking into account also other aerosol compounds. The Seifert and Beheng (2006) cloud microphysical scheme considers both phases. Also mixed-phase clouds are affected by the revision of the CCN parameterisation, e.g. via the Bergeron-Findeisen process. Nevertheless, the current analysis focuses on the liquid phase only, investigations on mixed and ice-phase clouds are left for future research.

## 2.2 Model evaluation method

Satellite retrievals have been used to evaluate the performance of the numerous GCMs and NWP models (e.q., Quaas et al., 2004, 2009; Zhang et al., 2005; Brunke et al., 2010; Cherian et al., 2012; Nam et al., 2014). However, a meaningful evaluation of modeling with satellite observations is challenging because of the difference in the model variables and the satellite retrievals. To address this problem, the integrated satellite simulator COSP (CFMIP Observational Simulator Package, Bodas-Salcedo et al., 2011) has been developed within the framework of the Cloud Feedback Model Intercomparison Project (CFMIP). The COSP satellite simulator produces model diagnostics, which are fully consistent with satellite products such as the International Satellite Cloud Climatology Project (ISCCP; Rossow and Schiffer, 1999), MODerate Resolution Imaging Spectroradiometer (MODIS; Platnick et al., 2003; Pincus et al., 2012), Cloud-Aerosol Lidar and Infrared Pathfinder Satellite

30  Observations (CALIPSO; Chepfer et al., 2010) and the CloudSat cloud radar (Marchand et al., 2009). To produce similar output to satellite data, COSP requires the grid mean vertical profile of temperature, humidity, hydrometer mixing ratios, cloud optical thickness and emissivity, the surface temperature and emissivity from the model. It produces output comparable with satellite data in three steps. First, it addresses the mismatch between model and satellite pixel resolution by generating sub-columns using model information about subgrid-scale variability e.g. from the assumption on vertical overlap of fractional cloudiness,

second, vertical profiles of individual sub-columns are passed to each instrument simulator, and third the COSP statistic module gathers the output from all instrument simulators (Bodas-Salcedo et al., 2011). COSP is implemented online in the COSMO model with hourly output. While using COSP facilitates a more consistent comparison between model output and satellite data, differences between the model simulation and the satellite can for example still arise due to displacements in simulated storm tracks. COSP has previously been used with COSMO by Mühlbauer et al. (2014, 2015). The output diagnostics include

a variety of cloud properties, which facilitate consistent model-to-observation comparisons as well as consistent inter-model comparisons.

In the next section, we evaluate the model results (derived using MODIS and the ISCCP satellite simulators) with MODIS level-2 and ISCCP satellite observations, in terms of cloud optical and microphysical properties (cloud optical depth, effective radius, liquid water path and cloud fraction). The MODIS satellite simulator uses profiles of particle size for liquid and ice

and corresponding optical depths within each layer of sub-column as a function of model levels. Using the cloud overlap assumption, zero or one cloudiness in each sub-column is created in each level. The diagnostics are then integrated over the cloudy sub-columns to obtain in-cloud average cloud optical depth and liquid water path. In turn, cloud effective radius is sampled at the cloud tops, which is not a vertical integral. Further, the ISCCP simulator aggregates pixel scale cloud retrievals (fraction of the sub-column with $\tau \geq 0.3$) to estimate cloud fraction (more details: Pincus et al., 2012).

The precision of weather forcasts for longer times is inherently limited by the non-linear nature of the problem. As forecast progresses, the uncertainty in weather prediction also increases. In turn, the earliest forecast time-steps are still substantially affected by the initialization. Hence, we have considered the third day of the simulation for evaluating the model with satellite observations. The synoptic condition which is discussed in the next section. To compare with model simulations, different swath data-sets of MODIS level-2 on 17 February 2007 (daytime overpass only) are combined and gridded to the model domain and

model outputs are averaged between 8.00-14.00 UTC (corresponding approximately to the MODIS-Terra overpass time over the domain). Also, MODIS level-2 products and model simulations are screened for liquid phase clouds because in COSMO-MUSCAT only cloud microphysics for liquid clouds was modified. Additionally, MODIS cloud optical depth and effective radius are applied with threshold values of 5 and $2\mu m$ (Sourdeval et al., 2016; Zhang et al., 2012a). Since the analysis is carried out for winter, satellite retrievals can be affected by snow cover on the ground. However, the MODIS retrieval (Platnick

et al., 2001) uses a combination of absorbing spectral channels for which the snow/ice albedo is relatively small which makes it suitable for retrieving cloud properties over snow. Furthermore, the COSP diagnosed model clouds are compared to ISCCP daily cloud products. For that, modeled ISCCP cloud products are re-gridded from 28 km to 280 km resolution (ISCCP satellite resolution) using a grid interpolation method and daily averaged. Besides these satellite observations, Clouds and the Earth's Radiant Energy System, (CERES, Loeb et al., 2012) satellite observations are also used for model evaluation, which are daily

products (Kato and Loeb, 2003). One should keep in mind that the satellite products, just like models, are prone to biases. Nonetheless, the spatial correlation of the cloud structures is well represented (Noble and Hudson, 2015; Min et al., 2012).

### 2.2.1 Numerical Simulations

To isolate and analyze the effects of the model modifications, we have performed three different model simulations with the same interpolated GME initial conditions for the time period of 10 days (15 - 24 February 2007). They are (a) a standard COSMO two-moment simulation with fixed $C_{ccn}$ ($3.0 \times 10^8 m^{-3}$) (COSMO2M), (b) a COSMO two-moment simulation with radiation coupled to cloud microphysics (COSMO2MR) which uses equations 3 to 5 in the radiation scheme (here also $C_{ccn}$ is kept fixed as in COSMO2M), and (c) a coupled COSMO-MUSCAT simulation, i.e. using interactive rather than prescribed $C_{ccn}$ and treating radiation in the same way as COSMO2MR (COSMO-MUSCAT). In most of the discussion we have used simulations (a) and (c). In all three model versions (COSMO2M, COSMO2MR, and COSMO-MUSCAT), we make use of the COSP diagnostics for the MODIS and ISCCP satellite simulators.

## 3 Results

### 3.1 Synoptic situation

The simulation starts on 15 February and ends on 24 February 2007. At the beginning of the simulation (00:00 UTC), the meteorological condition is dominated by a low-pressure system over the north Atlantic and a high-pressure system over land. The 2-m temperature shows a temperature gradient with a warm ocean and a cool continent, mostly in the northeastern part of the domain. The winds are mostly strong southwesterly over the Atlantic and northerly and northwesterly in the southern region (Figure S1). Since the case study has been conducted for 17 February, the model derived key meteorological parameters at 12:00 UTC are illustrated in Figure 2. On February 17, the low-pressure system has moved to the French Atlantic coast and a cyclonic circulation has set up over the region. Furthermore, a strong high pressure is seen over northeastern Europe. The 2-m temperature shows that prominent winter synoptic conditions still exist in the northern part with a warm oceanic region (Atlantic) and cold northeastern part. The southern region has a maximum temperature of $20°C$, whereas the northeastern continental region experiences a minimum temperature of $-20°C$. The cyclonic circulation drives the airmass from the oceanic region and results in the formation of clouds along the frontal systems. Besides, the high pressure in the Eastern part of the domain results in a cloud-free region due to subsidence. However, most of the domain is cloud covered with cloud fraction close to 100%. Furthermore, rainfall around 100 mm (accumulated precipitation over 96 h) on 17 February is observed along with the cyclonic circulation and the south eastern part of Europe, south of the low pressure system. The modeled convective cloud bases are located between 500 to 4000 m over the domain.

## 3.2 Evaluation with satellite data

The model derived cloud fraction is daily averaged (0:00 to 24:00 UTC) to illustrate the comparison between the model (COSP) and ISCCP satellite retrievals (Figure 3). The observed cloud fraction shows more cloud-free regions compared to the model simulations. Nevertheless, the model derived cloud fraction is in broad agreement with ISCCP satellite retrievals, allowing now for a more detailed analysis of the cloud microphysical properties with a fine resolution which is the center of this study.

Furthermore, a comparison of radiative fluxes with CERES (Clouds and the Earth's Radiant Energy System) satellite products is discussed in section 3.3.

Figure 4 shows the comparison between MODIS retrieved (Figure 4a-c) and COSMO-MUSCAT simulated (Figure 4d-f) cloud optical depth, cloud droplet effective radius, and cloud liquid water path. From the figure, it can be noticed that the simulated cloud optical depth exhibits a spatial pattern similar to the observations, albeit higher magnitude in the MODIS

level-2 retrievals (Figure 4a and d). In the satellite retrievals, it varies between 5 to 100 and in the model between 5 to 45, with maximum values observed over similar geographical regions. However, the satellite-derived cloud optical depth and liquid water path are larger in comparison with model (COSMO-MUSCAT) outputs. The model derived cloud effective radius exhibits both a similar spatial pattern and magnitude compared to that of the MODIS satellite retrievals (Figure 4b and e). The modeled cloud droplet effective radius varies between 3 to 16 $\mu$m, whereas it is in the range between 2 to 30 $\mu$m in the

satellite retrievals. Similar to cloud optical depth, liquid water path also exhibits comparable spatial patterns for both, model and observations. Whereas the modeled liquid water path varies between 0.025 and 0.425 $kgm^{-2}$, and in the satellite observation it varies between 0.25 and 1.0 $kgm^{-2}$. The white regions region (missing values) in satellite retrievals can be explained by the very strict quality-filtering of the MODIS cloud products. The domain averaged cloud optical depth, effective radius and liquid water path are 23.34, 11.30 $\mu$m, 0.175 kg$m^{-2}$, whereas the COSMO-MUSCAT derived values are 7.60, 9.93 $\mu$m, and

0.056 kg$m^{-2}$, which illustrates an underestimation of all simulated quantities compared to the satellite derived cloud optical properties. The above cloud optical properties are calculated using equations 3 and 5 in the models, although their correlations are valid only for that particular model layer/levels.

In the following, the statistical distribution of satellite and the model cloud microphysical properties are compared and evaluated in terms of probability density functions (PDFs). Figure 5 represents the probability density function of the spatiotemporal

distribution of cloud optical depth, effective radius, and liquid water path, defined as the normalized count of occurrence per bin width of cloud optical property. The cloud optical depth PDF shows that thin clouds (cloud optical depth $<$ 10) in all model versions occur substantially more frequently than in the satellite retrievals, and thick clouds (cloud optical depth $>$ 30), less frequently. The modeled cloud effective radius PDFs is constrained to 3 and 16 $\mu$m, where as the satellite retrievals shows a range of 4 to 30 $\mu$m. A shift of the PDF is found in the COSMO-MUSCAT derived PDFs, which indicates the increased droplet

size for the interactive $C_{ccn}$. For liquid water path, modeled PDFs overestimates the clouds with low liquid water path and underestimates clouds with high water paths. The differences in PDFs largely follow what is found for the cloud optical depth, but model deficiencies compared to the satellite retrievals are substantially larger. The analysis also illustrates an increased in cloud optical PDF from COSMO-MUSCAT simulation. Certainly, the drop and preponderance of modeled cloud optical properties

can be influenced by model tuning, an approach which, however, hasn't been performed yet for the COSMO-MUSCAT model version.

The outcome of the cloud microphysics modifications can be analyzed by considering the difference between the two simulations (COSMO-MUSCAT and COSMO2M), which is shown in Figure 4g-i. The version considering the interactive aerosol number concentration (COSMO-MUSCAT) exhibits an increase in the cloud droplet effective radius by a range of 1-4 $\mu$m throughout the domain with an overall increase of 9.5%, although a slight reduction can be noticed in a few areas. For cloud optical depth and liquid water path, both generally show increases despite the reduction in a few areas. The revised parameterization in the coupled model has modified the spatial distribution of the cloud optical depth in the range of $\pm$ 15 and the liquid water exhibits a variation in the range of $\pm$ 0.12 $kgm^{-2}$. However, the domain averaged cloud optical depth and liquid water path has been increased by 4.1% and 14.2%, which is also observed in PDF analysis.

The cloud droplet number concentration $N_d$ can be used as a diagnostic for aerosol cloud interaction. From satellite observations it can be expressed in terms of cloud optical depth $\tau_c$ and effective radius $r_e$ (Quaas et al., 2006), which is given by:

$$N_d = \alpha \tau_c^{0.5} r_e^{-2.5} \tag{9}$$

where $\alpha = 1.37 \times 10^{-5} m^{-0.5}$. Likewise, the model derived $N_d$ is also estimated using equation 9, which uses COSP (MODIS simulator) derived cloud optical depth and effective radius. Figure 6 (a-c) shows the comparison between modeled (CSOMO2M and COSMO-MUSCAT) and observed $N_d$. On 17 February 2007, the domain-averaged $N_d$ values are 153, 120, and 378 cm$^{-3}$ for COSMO2M, COMSO-MUSCAT and MODIS, which indicates an underestimation of model derived values (Figure 6a-c) compared to MODIS (Zhang et al., 2012b; Storelvmo et al., 2009). The inter-model comparison (COSMO2M and COSMO-MUSCAT) reveals that COMSO-MUSCAT derived $N_d$ shows a decrease of 21.5%. Figure 6d shows the spatial distribution of sulfate aerosol number concentration (aerosol number concentration proxy) below the convective cloud base (representative of aerosols in the model and it is also averaged for 8-14 hours on 17 February 2007), where high number concentrations are simulated over southeastern Europe. On contrast, $N_d$ are smaller over the same region. This is because the Boucher and Lohmann (1995) parameterization models saturation of $N_d$ over high aerosol or polluted regions (Penner et al., 2001) and the high pressure in this region results in trapping aerosol in the boundary layer. From the above analysis, it can be inferred that COSMO-MUSCAT can be used as a tool for regional aerosol-cloud interaction estimates. The interactive aerosol coupling results show an increase in cloud droplet effective radius by 9.5% and a reduction in $N_d$ by 21.5%.

### 3.3 Impact on radiative balance

In addition, we have also implemented aerosol-cloud-radiation interactions in the COSMO model by revising the radiation scheme to make use of a droplet-size-dependent cloud optical depth. Incorporating aerosol-cloud-radiation interactions in the model causes significant change in the radiation fluxes. Figure 7 shows the spatial distribution of net downward shortwave flux at the surface, net downward longwave flux at top of the atmosphere (TOA), and the corresponding difference between COSMO2M and COSMO2MR simulations (with fixed $C_{ccn}$). Similar to the above analysis, we also compare fluxes for 17

February 2007, and are daily averaged. The figure shows that in the radiation modified version (COSMO2MR) there is an increase in the net downward shortwave flux at the surface. Likewise, an overall reduction is observed in the net downward longwave flux at the TOA despite the increase in the northeast part of the domain. The net downward shortwave radiation at the surface shows an increase of about 10 to 60 $Wm^{-2}$ and the net downward longwave flux shows a decrease of 10 to 40 $Wm^{-2}$. This illustrates the reduction of cloud cover in the COSMO2MR simulations, which implies that reduced cloud cover results in more shortwave radiation reaching the surface and less longwave radiation reflected back to TOA. This can be inferred by
considering the cloud optical depth, and liquid water path difference between the COSMO2M and COSMO2MR simulations, which is also daily averaged (Figure 8). From the figure, the regions with reduced cloud optical depth and liquid water path are correlated with increased net shortwave flux at the surface and decreased net longwave flux at the TOA. To illustrate the combined effect of revised radiation scheme and interactive aerosols, COSMO-MUSCAT derived radiative fluxes are compared with CERES satellite observations (Kato and Loeb, 2003; Loeb et al., 2012). For comparison, we have considered computed
CERES fluxes (derived based on state and composition of the atmosphere, surface, and the incoming solar radiation) with spatial resolution $1° \times 1°$ and care must be taken while interpreting the results. Also, during winter the uncertainty in CERES flux observation are slightly higher (Guo et al., 2007). The spatial pattern and the magnitude of model simulated fluxes are comparable with satellite observations, in which the surface net downward shortwave flux ranges between 20 to 26 $Wm^{-2}$ and TOA net downward longwave flux varies between -290 to -140 $Wm^{-2}$ (Figure 9). Additionally, correlations between satellites
and models (COSMO2M and COMO-MUSCAT, model outputs are re-gridded to satellite resolution) are illustrated in Figure 9 c and d. The models modifications (revised radiation scheme and interactive aerosols) result in an increase in the correlation coefficient from 0.61 (COSMO2M) to 0.84 (COSMO-MUSCAT) in the case of net shortwave flux at the surface, whereas the modifiations do not have much effect on the longwave flux.

## 4    Conclusions

This paper discusses the modification of the Seifert and Beheng (2006) two-moment scheme in COSMO model. This has been done with aerosol information from the online-coupled MUSCAT model, which allows for a microphysical aerosol effect on clouds. It has been achieved by replacing the constant cloud condensation nuclei profile in the COSMO two-moment scheme with gridded aerosol information derived from online-coupled MUSCAT model, using the Boucher and Lohmann (1995) parameterization, which takes sulfate aerosol as a proxy for all aerosols. In addition, the radiation scheme is revised
to a droplet-size-dependent cloud optical depth, allowing now for aerosol-cloud-radiation interactions. In order to facilitate an evaluation using satellite retrievals, the COSP satellite simulator has been incorporated into the modeling system, which runs online in the model. The model results are evaluated with satellite observations from the ISCCP, MODIS, and CERES projects and instruments, respectively. Since the cloud microphysics modification has been done for cloud droplet nucleation, the analysis are restricted to the liquid part of clouds in the model and MODIS level-2 cloud products are screened for liquid phase
cloud products. Although the two-moment cloud microphysics and radiation scheme in COSMO model has been modified, the model was not re-tuned to get reasonable 2m temperature or precipitation.

A case study has been carried out to compare the model output with observations. The incorporated COSP satellite simulator serves as a link between model and satellite comparisons. Despite the resolution, COSP derived ISCCP cloud fraction shows similar spatial pattern and magnitude. Further, MODIS level-2 cloud optical products such as cloud optical depth, effective radius, and liquid water path are compared. The COSMO-MUSCAT derived cloud optical properties show a similar spatial distribution compared to the MODIS observation. In COSMO-MUSCAT, the cloud optical depth has been increased by 4.1%,

cloud droplet effective has been increased by 9.5%, and liquid water path has been increased by 14.2% in comparison to CSOMO2M. In turn, the cloud droplet number concentration estimated from COSMO-MUSCAT model shows a reduction of 21.5% compared to the COSMO2M model. Furthermore, considerable changes in the radiation budget have been found. This analysis indicates that the coupled model (COSMO-MUSCAT) with interactive aerosol treatment results in an increase in cloud droplet size and reduction in cloud droplet number concentration by activation and growth of droplets, which illustrates

implicit aerosol-cloud interactions. Also, the cloud properties in COSMO-MUSCAT agree reasonably well with observations, so that it can be used for regional aerosol-cloud interaction studies.

As a next step, further improvement in the two-moment scheme will be carried out through the use of the newly included aerosol model M7 (Vignati et al., 2004) framework in the MUSCAT model, which is able to provide aerosol number concentration information to the COSMO two-moment scheme by replacing Boucher and Lohmann (1995) parameterization. This can

result in more precise cloud droplet activation parameterization, involving different aerosol species as $C_{ccn}$, and thus improving microphysical aerosol effect on clouds (Lohmann et al., 2007). Also, the role of aerosols on ice nucleation will be addressed.

### Code and data availability

The COSMO-MUSCAT(5.0) model is freely available under public license policy. The source code, external parameters and

documentation can be obtained through Ralf Wolke (wolke@tropos.de).

*Acknowledgements.* This work was supported by an ERC starting grant "QUAERERE" (GA no 306284). We acknowledge the development work by the COSMO consortium. We thank Axel Seifert and anonymous reviewers for their valuable suggestions.

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

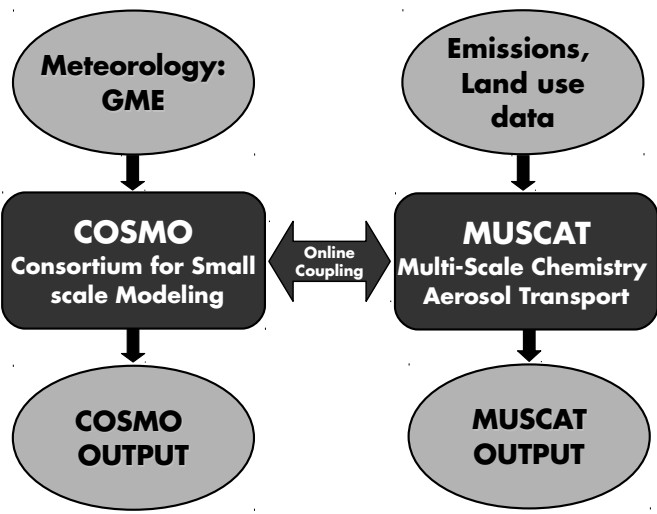

**Figure 1.** COSMO-MUSCAT modeling system. Left hand side, setup of COSMO modeling system with GME input. Right hand side: MUSCAT modeling system with land use and emissions.

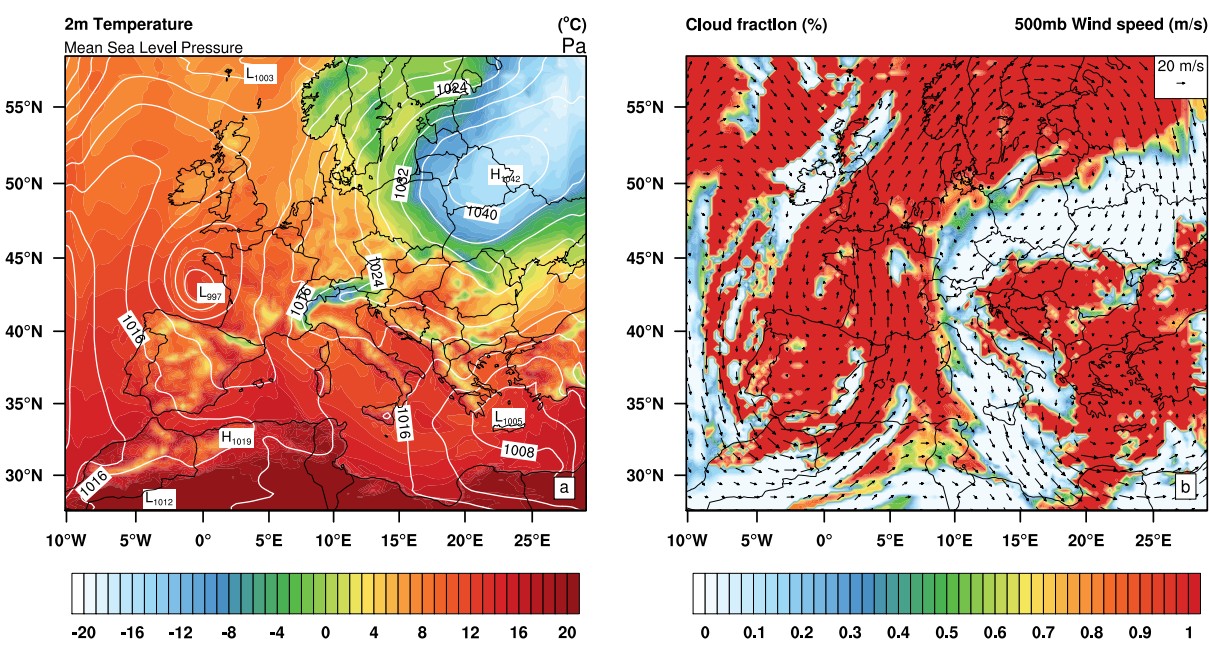

**Figure 2.** Model synoptic conditions for 17 February 2007 at 12:00 UTC, (a) Surface pressure in contours and 2 m temperature (°C) as colour shading, (b) 500 mb wind vectors and total cloud area fraction as colour shading.

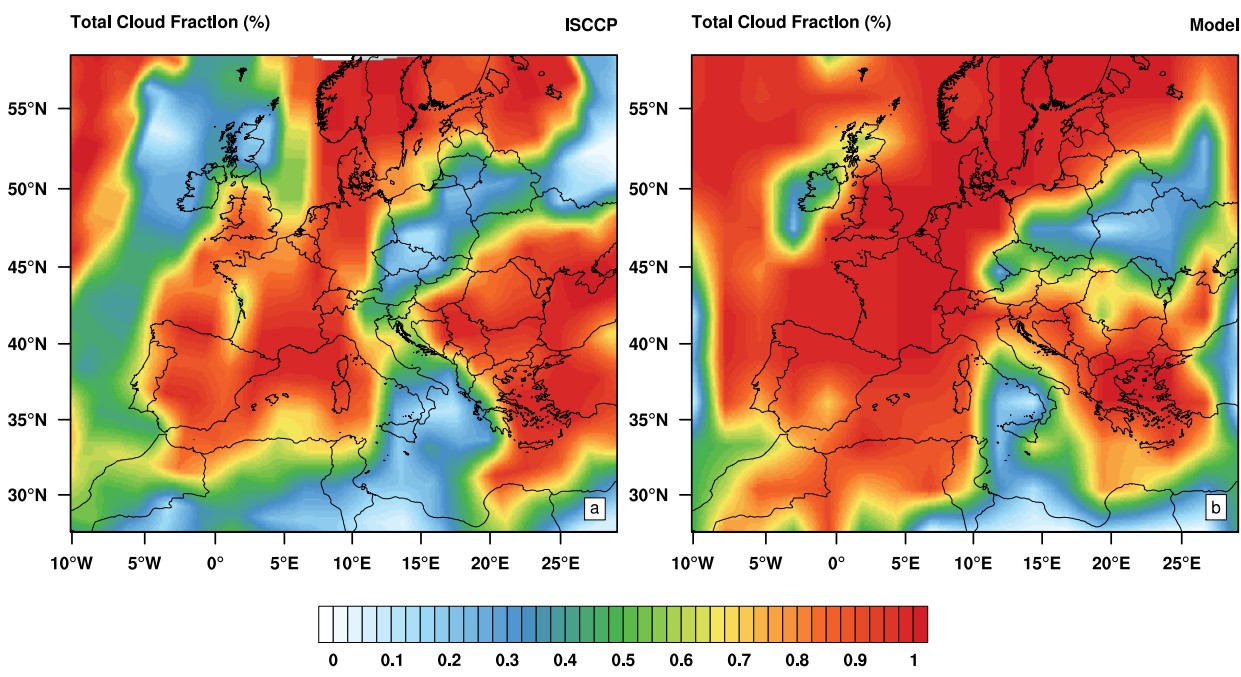

**Figure 3.** (a) Satellite and (b) model (COSMO-MUSCAT) derived ISCCP cloud fraction, for 17 February 2007 (daily averaged).

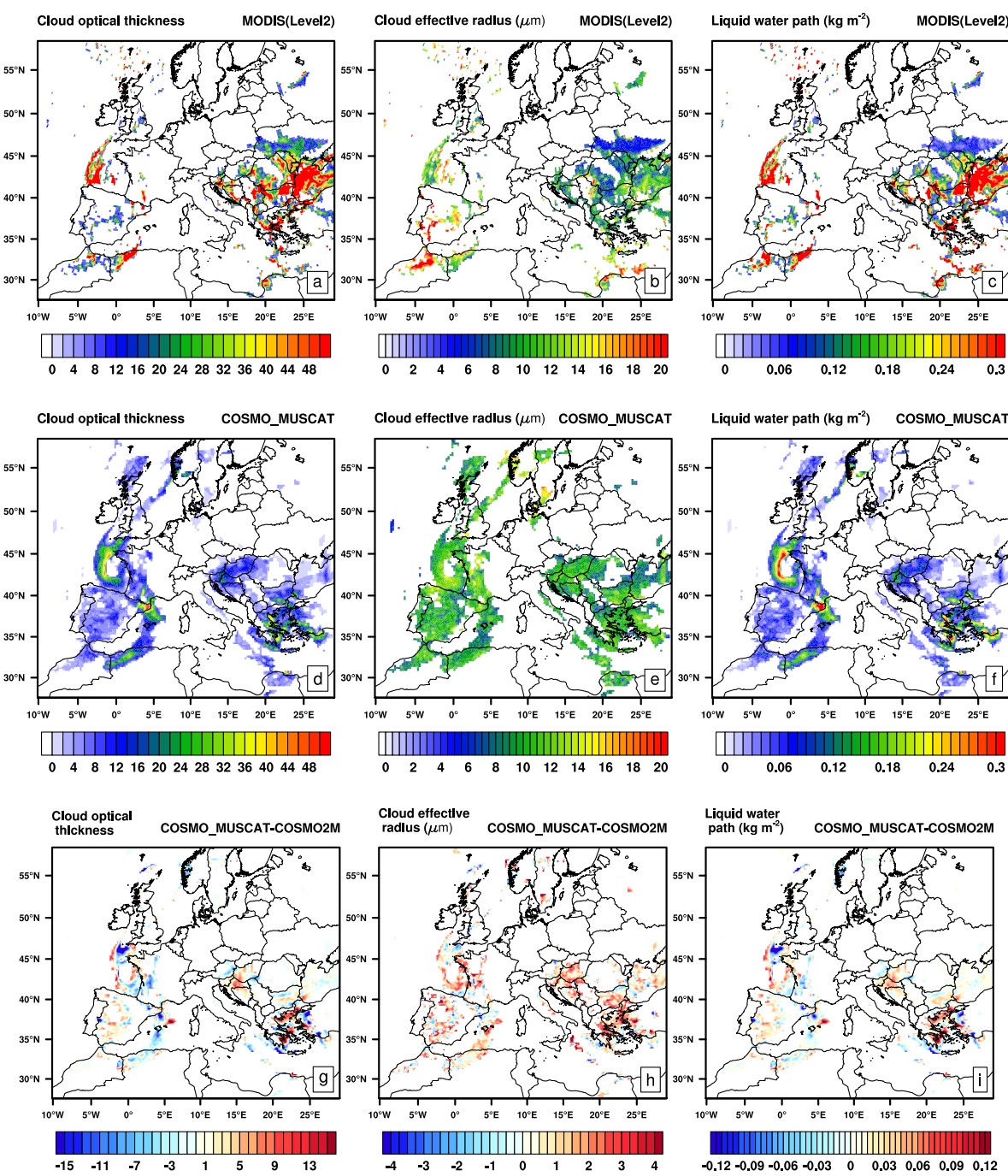

**Figure 4.** MODIS Level-2 (a) cloud optical depth, (b) cloud effective radius, (c) cloud water path, COSMO-MUSCAT derived (averaged between 8.00 -14.00 UTC, approximate MODIS-Terra overpass time over the domain) (d) cloud optical depth, (e) cloud effective radius, (f) cloud water path, and difference between COSMO-MUSCAT and COSMO2M simulations (g,h,i), for 17 February 2007.

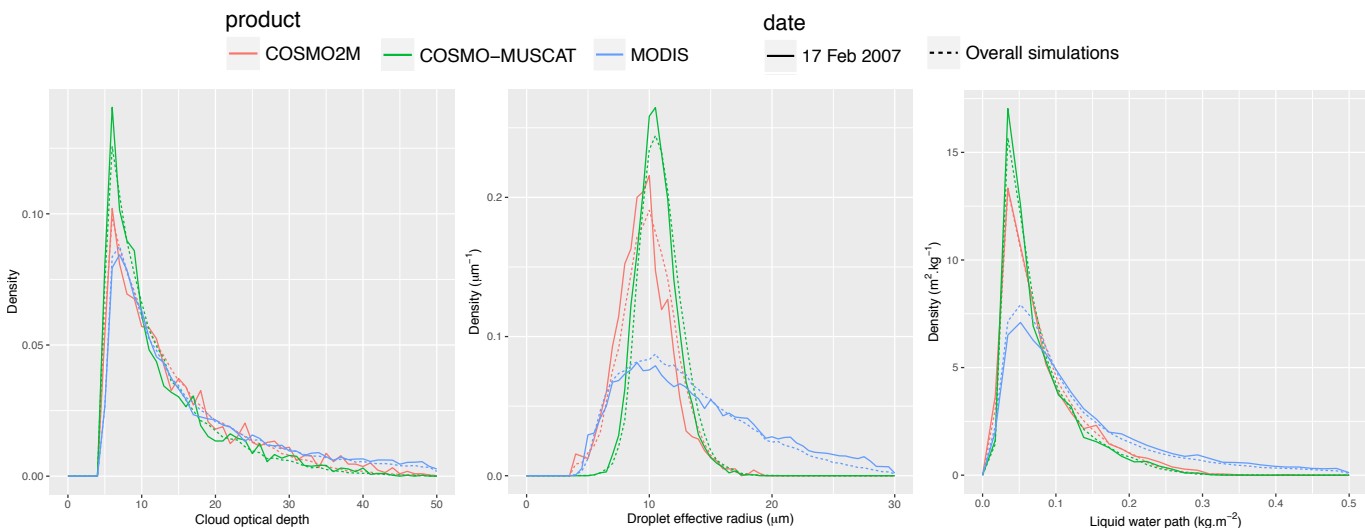

**Figure 5.** Probability density functions (PDF) of cloud optical depth, cloud effective radius, Liquid water path from COSMO-MUSCAT (green), COSMO2M (red) and MODIS Level-2 products (blue), for 17 February 2007 (solid line) and for 10 day period (15-24 February 2007) simulation (dashed line).

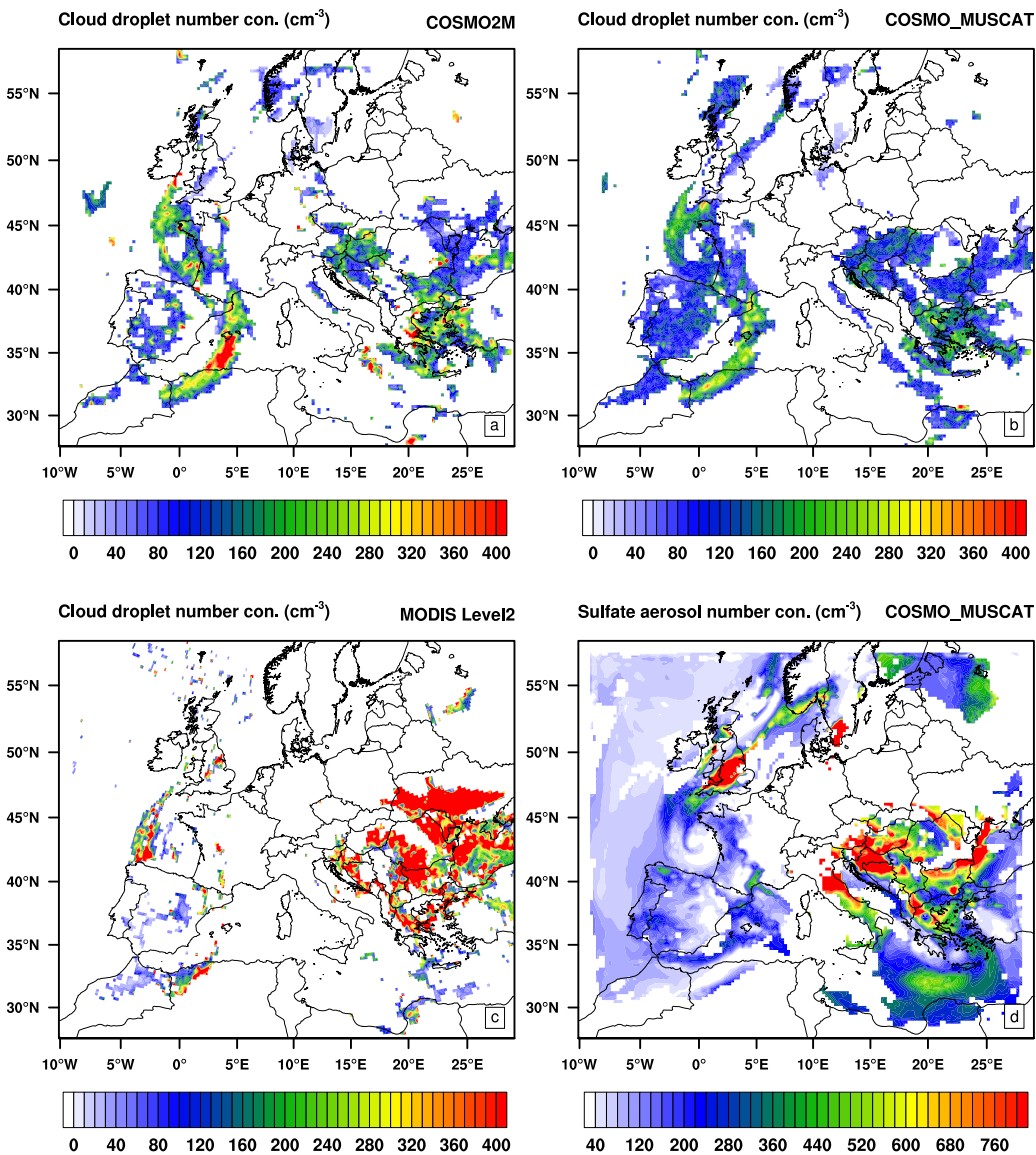

**Figure 6.** Cloud droplet number concentration (averaged between 8.00 -14.00 UTC, MODIS-Terra overpass time over the domain) for (a) COSMO-2M, (b) COSMO-MUSCAT, (c)MODIS level-2 , and (d) Sulfate aerosol number concentration below the convective cloud base from MUSCAT model, for 17 February 2007.

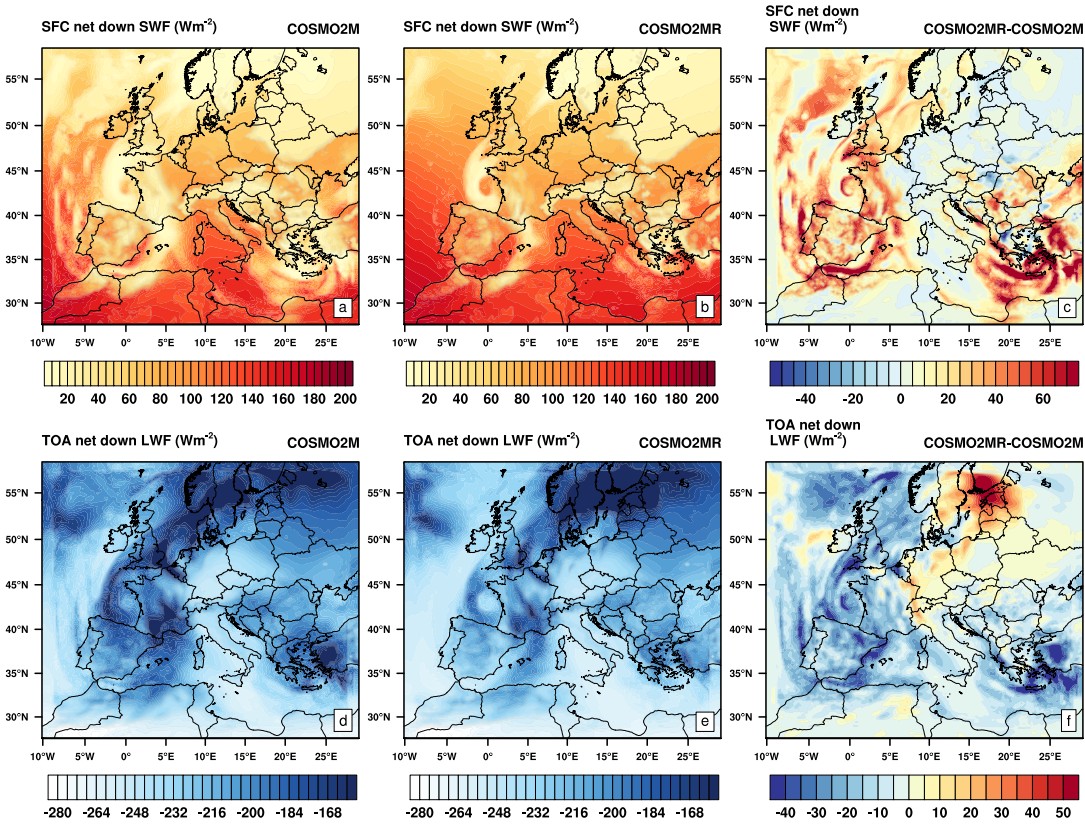

**Figure 7.** Comparison and difference between short wave and long wave radiation fluxes surface and top of the atmosphere, and it is difference between two simulation (COSMO2MR radiation coupled minus COSMO2M).

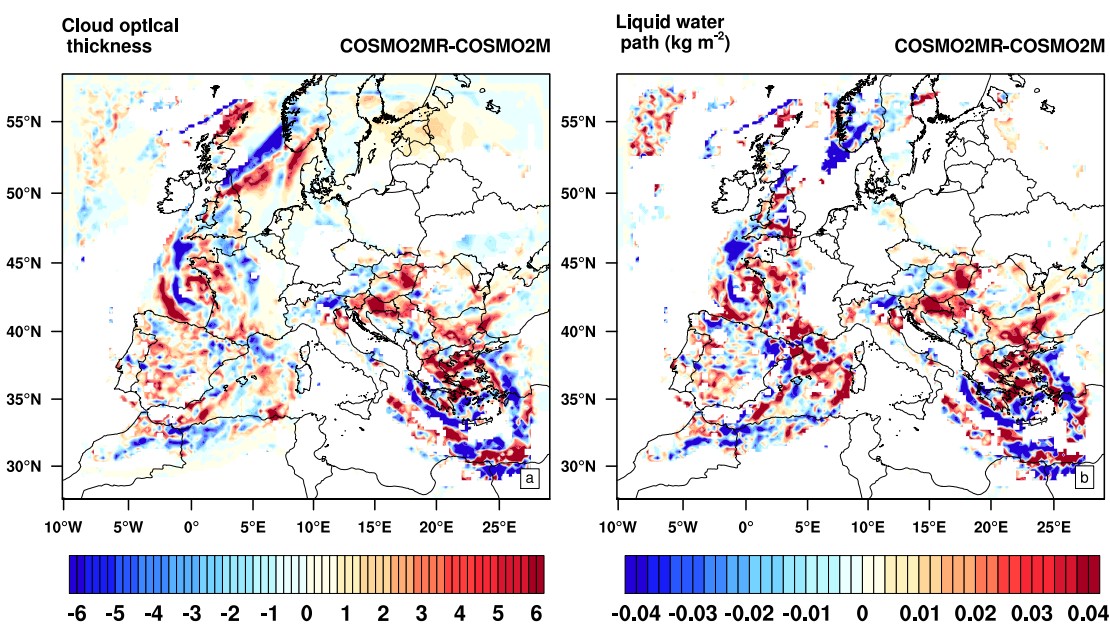

**Figure 8.** Daily averaged cloud optical depth and liquid water path difference between COSMO2MR and COSMO2M on 17 February 2007.

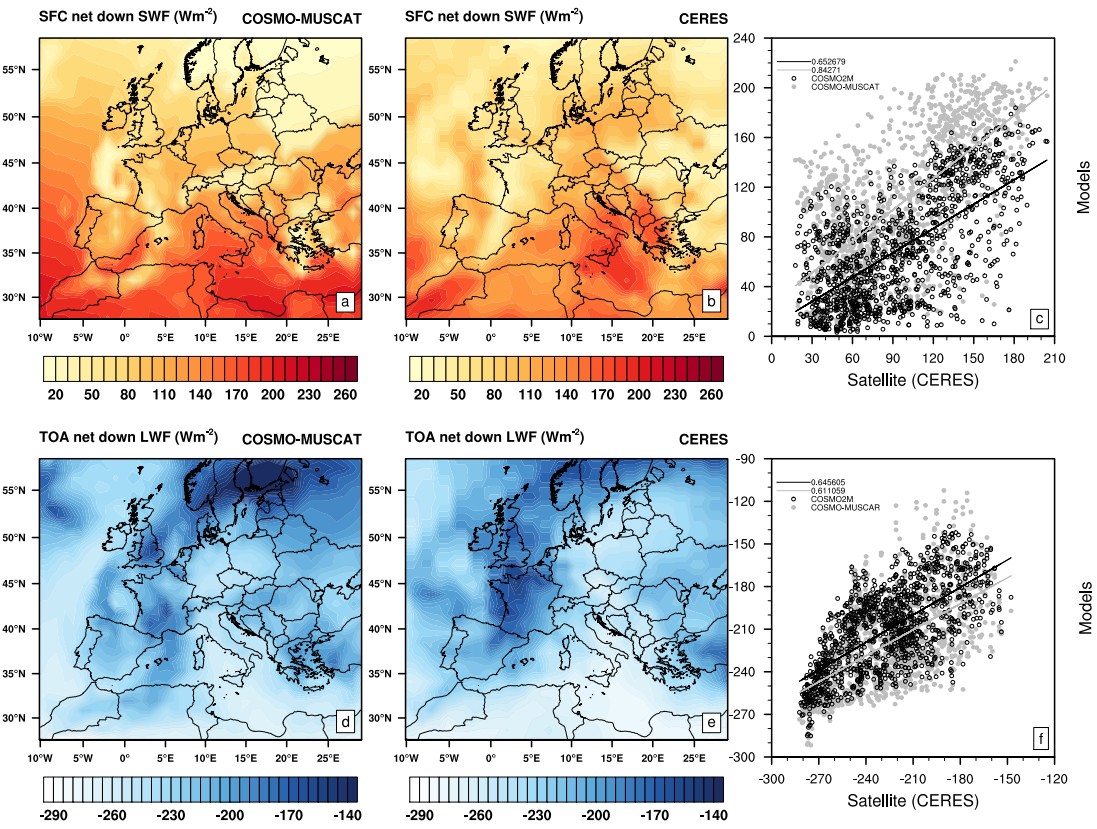

**Figure 9.** Comparison between short wave and long wave fluxes at surface and top of the atmosphere with CERES satellite fluxes and correlation between satellite (CERES) and models (COSMO2MR and COSMO2M).

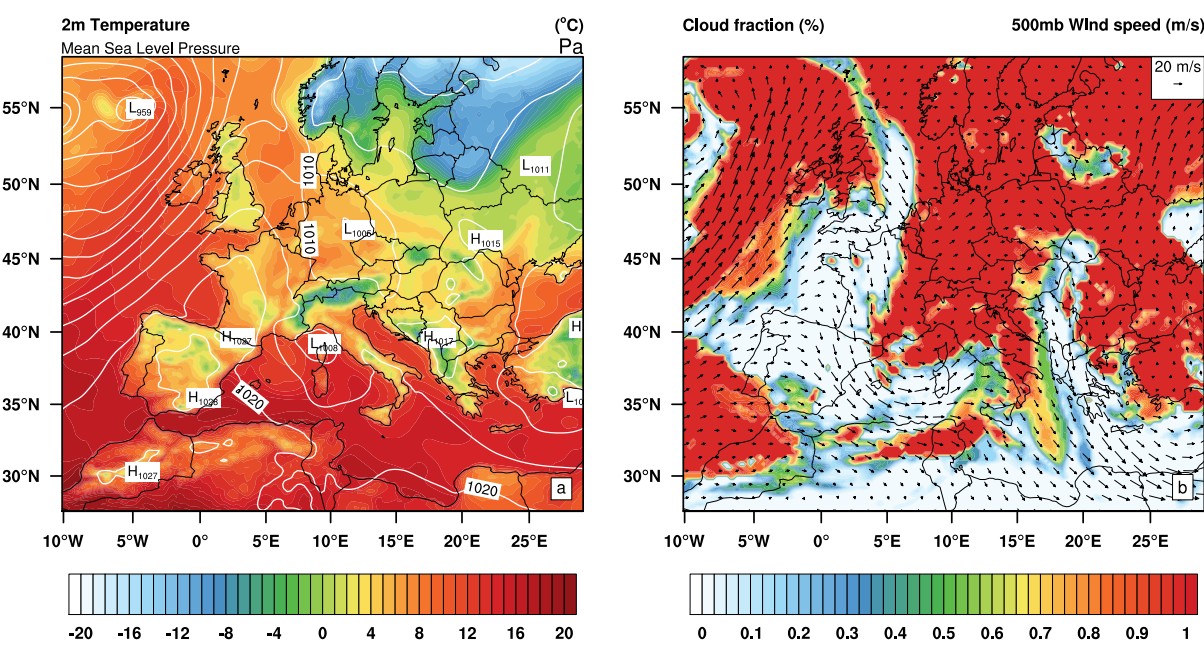

**Figure 10. Figure S1:** Model synoptic conditions for 15 February 2007 at 00:00 UTC, (a) Surface pressure in contours and 2m (°C) as colour shading, (b) 500 mb wind vectors and total cloud area fraction as colour shading.