# Peer review of "Implementation of aerosol-cloud interactions in the regional atmosphere-aerosol model COSMO-MUSCAT(5.0) and evaluation using satellite data"

_Geoscientific Model Development, 2016_

## Short Comment (SC1) · 13 Sep 2016

Dear authors,

In my role as Executive editor of GMD, I would like to bring to your attention our Editorial version 1.1:

http://www.geosci-model-dev.net/8/3487/2015/gmd-8-3487-2015.html

This highlights some requirements of papers published in GMD, which is also available on the GMD website in the 'Manuscript Types' section:

[Figure]

http://www.geoscientific-model-development.net/submission/manuscript_types.html

In particular, please note that for your paper, the following requirement has not been met in the Discussions paper:

- "The main paper must give the model name and version number (or other unique identifier) in the title."

Please add the version numbers for COSMO and MUSCAT or for COSMO-MUSCAT (depending on your versioning system) in the title upon your revised submission to GMD.

Yours,

Astrid Kerkweg

---

## Referee Comment (RC1) · Anonymous Referee #1 · 20 Sep 2016

The authors present two extensions to the regional atmosphere aerosol chemistry model COSMO-MUSCAT. As a first extension, the authors now use the two-moment scheme from COSMO (Seifert and Beheng). The cloud condensation nuclei (CCN) information needed by this scheme comes from MUSCAT (instead of constant, prescribed CCN profiles), following Boucher and Lohmann 1995 and taking sulfate mass as a CCN proxy. The second extension concerns the cloud optical depth in the radiation scheme, which now accounts for droplet-size, via the cloud effective radius, following Martin et al. 1994.

To evaluate the effect of the two new features on code performance, the authors consider a ten day test period (February 15 to 25, 2007). Of this period they focus, however, mostly on one single day (February 17). The simulations are run in forecast mode, thus are not nudged. Variables considered for the evaluation are different cloud related quantities (cover, optical thickness, effective radius, water path, droplet number concentration) and shortwave and longwave net radiation at the surface and the top of atmosphere (TOA). Comparison is done among different model versions (COSMO2M, COSMO2M.rad, COSMO-MUSCAT) and with satellite data (CERES, MODIS, ISCCP) and typically comes in the form of maps.

The authors find significantly improved performance of the new code version when comparing modeled and satellite based cloud effective radius and cloud droplet number concentration. Improvements are less pronounced for other quantities like cloud optical depth, cloud water content, or cloud fraction.

The topic - effect of more elaborated aerosol-cloud treatment in a regional climate model - is of interest. A number of corresponding models exists (e.g. WRF or also COSMO, Zubler et al. 2011), yet given the complexity of the topic a larger number of models whose results can then be compared are clearly desirable. The study thus is of interest.

The study fits the scope of GMD, but requires major revisions to meet GMD standards.

**Major points:**

1) Precision and / or clarity of statements could generally be improved. Two examples in the following, more can be found under 'minor points' below.

Evaluation is currently done essentially via comparing maps and arguing that things look similar or that there is a slight increase, minor change, a largest change etc. What are the numbers behind such statements? Only few are given. Regional averages, variability, correlations, scatter plots etc. would allow for better quantitative comparison

of the different models among themselves and with the satellite data.

What is the resolution (space and time) of the satellite data you use for evaluation? Are model averages based on data from each model time step or based on output data? If output: hourly or less frequent?

2) You state that you have run the model for 10 days in forecast mode, February 15 to 25, 2007. But you use only one day (February 17) for model evaluation. Why? More importantly, is a one day forecast enough to evaluate the different models? The model may still, above all, be in an adapting stage after only one day (see e.g. Cossu Hocke, GMD, 2014). This may also apply to aerosols, a key element in your study, but with lifetimes on the order of days. While the forecast mode of your simulations makes comparisons more difficult as time evolves, you may still check whether, for example, CCN and cloud optical thickness evolve in concert over the ten days of your simulation. Please comment on possibilities and limitations of your one day forecast comparison. Or re-run your simulations in nudged mode an compare them over a longer time.

**Minor points:**

p.2, l.30-35: As you point out that other groups have already coupled COSMO with two-moment cloud microphysics to an aerosol module, including droplet-size aware radiation. Please explain to the reader how your work differs from these existing, closely related approaches.

Section 2.1 I find rather difficult to read as information on different codes (COSMO, COSMO-MUSCAT, and MUSCAT) as well as different code versions (current version, several other versions) is tightly interleaved. It is not always obvious whether what is stated applies to COSMO, COSMO-MUSCAT, the former or present code etc. For example, p.4, l.23ff: to which model do these equations now apply? COSMO2M? If so, how do the CCN numbers given here (1.26e9 and 1.0e8) go together with the 300 mentioned p.8,l.6? I further guess, but this is not really clear from the text, that equation 7 also applies to COSMO-MUSCAT, but with Cccn taken from MUSCAT and probably

k and Smax the same as in COSMO. Please clarify.

p.4, eq.3: What is Gamma?

p.5,l.3: 'aerosol mass concentration information from the MUSCAT model'. Where does MUSCAT get that information from? Are aerosol emissions prescribed? If so, where from? Or concentrations? What are these emissions / concentrations?

In section 3, evaluation against satellite products: is snow cover an issue?

p.6, l.6: 'with radiation coupled with microphysics': you mean the cloud effective radius following Martin et al. 1994 (your equations 3 to 5) is used? But for a fixed CCN number of 300?

p.6,l.27: Unless you have further evidence that for the concrete case ISCCP indeed underestimates cloud cover over the Atlantic, a fairer formulation may be that besides the model having a problem it could also be that the satellite has a problem.

p.7, l.2: 'In the two model versions...' You consider three model versions, COSMO2M, COSMO2M.rad, and COSMO-MUSCAT. While by and by one finds out what two versions you mean here, please state so explicitly.

p.7, l.3ff: Can you comment further on this screening for the liquid phase in MODIS and the models? How dominant is the liquid phase in either one?

p.7, l.9: 'In both cases it varies between 2 and 50...' Does the real quantity vary in that range or just your colormap? Also, figure 4a shows clearly much more red color than figure 4d. Reducing the comparison of the two panels to their range skips this aspect. In that sense, the patterns are not similar, as claimed in the text. Also, in wide parts where there is substantial cloud optical depth, the satellite based value is about twice as large as the model value. I would not call this a slight difference but a factor of two. This is another example where precision could be improved (major point 1).

p.7, l.14 ; High values for cloud effective radius are also seen over land. And, as above

(p.6, l.27), it is not obvious that the flaw is with the satellite data.

p.7, l.21: What do you mean by 'largest impact'? Largest in what sense? Change in mean? Median? Percent or absolute? Per grid box?

p.7, l.25: 'slight increase': what is slight?

p.7, l.26: 'The cloud optical depth shows a variation in the range of +/-20...' Variation over what? Spatially? Within a model? Please clarify.

p.8, l.8: There is a wide region (red in figure 5d) with CCN of at least 300, i.e., the prescribed CCN value in COSMO2M. Yet the CDNC in COSMO-MUSCAT is much lower than in COSMO2M also in this region. Why?

p.8, l.14: 'While comparing with high resolution MODIS satellite products...': These have not yet been introduced, I think.

Section 3.3.: It would be interesting to elaborate a bit more on radiation. For example, the differences between COMSMO2M and COSMO2M.rad seem to be larger over sea than over land. True? Do the large differences (more downward SW and upward LW at the surface especially over sea) go hand in hand with reduced cloud optical thickness? Change in cloud effective radius? Cloud cover? Regarding the comparison with CERES: what area means of CERES and models? Given that you look at February (little radiation, snow cover, short days) and a cloud cover close to 100 percent over wide regions: how reliable are CERES surface fluxes?

p.8, l.30: 'This paper presents an initial approach to the modification of Seifert and Beheng (2006) two-moment scheme in the COSMO model.' This is not true. Other groups have done this before, e.g. Zubler et al. (2011) whom you cite.

p.9,l.2: Maybe state that this parameterization takes sulfate mass as a proxy, it is not a full grown aerosol module like SALSA, M7, etc.

p.9, l.8: 'In terms of the cloud distributions, this modification has only a minor effect.'

Given that you compare the second day of forecast simulations in winter, this is not truly surprising. To what degree is this finding just due to large scale weather conditions / your initialization?

p.9, l.9: What means daily averaged when you consider only one day in the first place?

p.9, l.10: 'The modified model simulations are in broad agreement with satellite observations.' I would argue that all your simulations are in 'broad agreement'. However, you see some improvements (as you state) in your modified version.

p.9, l.15: '...only minor changes in terms of the radiation budget were found.' Looking at figure 6, i-l, I would not call these changes minor. In wide regions they are on the order of a factor of two.

Figure 1: Why does it say "M7 to be implemented"? In the reference you cite, Wolke et al. 2012, it is stated that M7 is implemented. Please explain. And if M7 does indeed not form part of your model version, remove it from Figure 1.

Figure 2: What is the data source?

Figure 3: Does cloud cover from COSMO2M, and possibly COSMO2M.rad, look similar to cloud cover from COSMO-MUSCAT?

Figure 4: While the figure is useful, some more quantitative comparisons would also be useful, e.g. area means, variability, scatter plots... For example, what is the are mean cloud water path in 4i? And how does it compare wit the area mean of 4f? Figure 4i looks as if there is above all a change in spatial distribution of the cloud water path, not of total cloud water path (area mean). The same question may be asked for figures 4g and 4h.

Figure 4: How different would the figure be if you were to compare COSMO-MUSCAT with COSMO2M.rad? Put differently, are the differences mainly due to the variable CCN or also to the size-aware radiation?

[Figure]

Figure 5c: Point out that this is not a MODIS product but a derived quantity. Also, why is there hardly anywhere a CDNC greater than 10? After all, there is cloud cover all over the place and CDNC=10 or smaller is very low.

Figure 6: On the western boundary of the domain, there seems to be a boundary effect. Can you comment?

Figure 6 j-l: Given that the color scale is saturated in wide regions in these plots, why not take it larger? Possibly even -40 to +40, as in 6i?

Figure 7, a-d: Same figures as in figure 6 e-h. No need for duplication. You may consider replacing these panels with corresponding ones from COSMO-MUSCAT, so one has all three models and CERES shown.

Figure 7 6: What time is shown? February 17? 24h mean? Color scale in the SFC SW plots could be reduced to maybe 200 instead of 260 to fully exploit the range of the color table.

It maybe worthwhile to point out somewhere that you only changed the model but did not (yet) re-tune it, e.g. to get reasonable 2m temperature or precipitation. You show that the different codes give, for example, different cloud optical properties. But this does not imply an overall better model performance.

The language needs brushing, there are a number of sentences that do not work on the language level. I give only two examples.

p.2, l.6/7: "Although regional models do not describe part of the large scale feedbacks which are included in GCMs, regional modeling allowing for an optimal compromise.

p.8, l.8/9: "From figure 5c, the maximum aerosol mass concentration observed over south eastern Europe, on the contrary Nd shows less.
* * *

---

## Referee Comment (RC2) · Anonymous Referee #2 · 21 Sep 2016

**Implementation of aerosol-cloud interactions in the regional atmosphere-aerosol model COSMO-MUSCAT and evaluation using satellite data**

In this study the spatial and temporal covariance of aerosol concentrations and cloud condensation nuclei (CCN) concentrations is introduced in the MultiScale Chemistry Aerosol Transport Model (MUSCAT).  For this purpose an empirical relationship between the sulfate mass and CCN concentrations, which was introduced by Boucher and Lohmann (1995), is implemented. Furthermore, the warm-phase cloud microphysics is coupled to the radiation scheme using a widely accepted empirical relationship between the cloud optical thickness, the effective radius and the vertically integrated liquid water content within an atmospheric layer.
Their new model version is evaluated against satellite retrievals of cloud microphysical and macrophysical properties, as well as radiative flux measurements at the top-of-the-atmosphere and at the surface using satellite simulators implemented in the model. The evaluation is performed for 1 day on Feb. 17 2007.

Given that uncertainties with respect to aerosol cloud interactions remain high, testing alternative approaches in treating these processes is of interest to the community. Therefore this paper could be published in GMD following major revisions of the manuscript.

**Major Revisions:**

- Whilst the paper gives a detailed introduction of the general benefits of regional models with online coupled aerosol-cloud, aerosol-radiation and aerosol-cloud-radiation interactions, such as WRF-CHEM (Grell et al. 2005), COSMO-ART (Bangert et al. 2011), COSMO-M7 (Zubler et al. 2011) and COSMO-ART/M7 (chapter 2 of: http://e-collection.library.ethz.ch/eserv/eth:48845/eth-48845-02.pdf), I would like to be given further motivation regarding the advantages of their particular approach and have it contrasted to existing approaches.

  My questions raised here, could guide such a discussion:
  1) Why did you use the Boucher and Lohmann's (1995) empirical relation, rather than implementing newer approaches using Koehler theory that have previously been applied in COSMO-ART/COSMO-M7 such as Abdul-Razzak & Gan (2000), or Nenes & Seinfeld (2003)? What is your justification for only considering sulfate and ignoring nitrate contributions to CCN?

  2) Whilst the relationship between cloud optical depth, effective radius and vertically integrated liquid water content is a commonly used diagnostic, it is wavelength independent. How did you deal with this issue when implementing this approach in a radiation scheme that computes the radiative transfer within 3 SW and 5 LW bands covering the entire wavelength spectrum?

- Furthermore, a short motivation should be included regarding the chosen case study. Are all the clouds shown boundary layer stratocumulus?

- The model evaluation is performed on a single day. The authors argue that the forecast skill decreases with increasing lead time. Could you obtain better agreement and therefore obtain a

longer evaluation period if you performed nudged simulations? Otherwise, a brief justification should be given that 1 day is a sufficiently long time period for your evaluation.

**General Comments:**

- This paper talks about the coupling of aerosol-cloud interactions and aerosol-cloud-radiation interactions. However, it should be stated clearly (already in the abstract) that these interactions are only included explicitly for warm-phase cloud processes.

- The paper often talks about changes to the COSMO model in general. However, the modifications are applied not to the COSMO model in general, but to the COSMO version of COSMO-MUSCAT. Whilst I agree with the authors that CCN variability and aerosol-cloud-radiation coupling is not provided in the officially released code versions, such developments have already been included in other COSMO versions (COSMO-ART, COSMO-M7, COSMO-ART/M7). They are therefore not new to the COSMO code itself.

- The paper is written in good scientific English. However, some sentences require re-writing (some of which are listed below). Furthermore, articles are missing in a few places.

**Specific Comments:**

P2L6: Last sentence needs rewriting.

P2L15: Sentence needs rewriting:"This approach, however,..."

P2L31-32: Delete last sentence of this paragraph. It is mis-leading as all of the mentioned aerosol-coupled regional models include a coupling of the cloud microphysics to the radiation.

P3L22:Sentence needs rewriting:"In this study, the COSMO model..."

Equation 5: Change 1.5/2 to ¾.

P4L23: Sentence needs rewriting: "IN the COSMO model, the aerosol..."

P5L1-11: This discussion needs some clarification: How is the Smax issue raised overcome? You raise the issue of ignoring updraft velocity in the Boucher and Lohmann formulation, however, is this then not introduced by dS/dz*w in Eq.7 (which admittedly is a very simplistic formulation of this relationship).

P5L20ff: Here you list all satellite products used for evaluation. Also include CERES here, as you use it later.

P5L27: I suggest to delete: In the upcoming… You have 1 ISCCP figure, 2 Modis and 1 CERES. That is not massively unbalanced.

P5L30: I suggest to delete last sentence of this paragraph.

P5L25-30: Please rephrase. The point regarding satellite biases could be formulated like: "One should keep in mind that the satellite products, just like models, are prone to biases. Comparisons of satellite retrievals with … have shown that…. Nonetheless, spatial correlation of the cloud structures are well represented."

P6L2: Sentence needs rewriting: "As the forecast time..."

P6L10: Results are only shown for 17th. Not 15th – 25th. Please include description of the cloud types of this domain (altitude, phase, average thickness, surface precipitation). When did you start your simulation, cause the 2nd day would be the 16th if the simulation were started in the morning of the 15th? Also, why are is the 16th not included in the analysis?

P6L22-25: First 3 sentences should go into section 2.1.1 (methods).

P7L1-4: First 3 sentences should go into section 2.1.1 (methods).

P7L6: "The top panel shows…." This should be in Fig. Caption.

P7L19: Please check units.

P7L25-30: In this discussion quantitative statements should be included. For instance, the area mean changes + variability could be determined. It would help determine the signal from the noise in figures g to i.

P8L14: First sentence of paragraph need rewriting/clarification.

P8L19: Maybe rephrase title, because aerosol-cloud radiation interactions were already discussed in previous section (cloud optical thickness). Suggestion: "Impacts on radiative balance"
In general this discussion is not very precise. It would help the discussion if you relate observed changes in e.g. the SW fluxes to the decrease in cloud optical thickness…

P8L30: Please rephrase. It is an initial approach to modify COSMO-MUSCAT only.

P9L7: Finding 1 should be removed as it is not discussed in the paper.

P9L13: Paragraph missing after point 2.

P9L22: Last sentence needs rewriting and one reference is missing.

P10ff: A few references need changing. Some paper titles use capital letters.
        IPCC reference is incomplete

Fig1: More detail should be given in the caption. I think this figure could be improved/clarified.
       Why are the "emissions/land use" and M7 stand-alone and not connected?
       The figure could be clearer regarding the structure of the code. Are RACM and MUSCAT separate modules? Will M7 not be embedded in MUSCAT?
        Is the output really completely separate?

Fig2: I personally would not include all "H" and "L" markes. I would simply mark the center of the dominant low pressure and high pressure systems.

Fig3/Fig4: Looking at these figures optically, I would arrive at different cloud fractions for Fig3b and Fig4d. Areas where the total cloud fraction of the model is 100%, the diagnosed optical thickness is 0, or just very small?

Does the COSP simulator include subgrid-scale cloud water? If not, is that justified for cloud optical thickness?

Fig4: Maybe regional/domain means + a measure of variability could be given to highlight the results more quantitatively?

Fig5: What is going on with the MODIS cloud droplet number estimate? Why is the spatial pattern so different than in Fig4a-c. Are these clouds just too shallow to obtain a good number estimate?

Fig6/Fig7: I find the color scale depicting the fluxes very misleading. I would suggest using a pure blue color scale for LW fluxes and pure red for SW fluxes.

---

## Referee Comment (RC3) · Anonymous Referee #3 · 17 Nov 2016

**Review of Implementation of aerosol-cloud interactions .....**

by S. Dipu Sudhakar et al.

The authors present a numerical study on aerosol, clouds and radiation with mutual interactions and compare the results with saltellite derived data. This is an important topic, since all cloud-related processes pose a severe problem in weather forecast and climate modelling. The paper contributes to the ongoing research by examining the effect of mutual interactions of these processes and the improvement of atmospheric models. This is worth to be published. The presentation is concise, length and number of figures are appropriate. However, the presentation is partly very vague and not consistent throughout the paper. The differences between the resp. data fields are inspected by eye, but not quantified. Therefore, it is difficult to follow the conclusions. Errors in some equations may be typos.

Yet, before publication, I suggest some substantial revisions. Please see the major points and specific remarks below.

**Major points**

• In midlatitude winter I expect that the ice phase plays an important role in the development of clouds and precipitation (Bergeron-Findeisen effect!), and you use the Seifert and Beheng (2006) scheme for mixed phase clouds. The paper, however, is devoted to the liquid phase alone.

Please discuss the effect of the modified treatment of drop nucleation on the ice phase properties, since a modification in one path of condensate formation is connected with an opposing trend in other path(s).

How do you determine the effective radius under cloud free conditions?

• Equations (3) - (5)

(3) holds if the cloud drop size distribution is used with the internal coordinate drop diameter D, not radius r. Then, (4) follows as

$$\lambda = \left[\frac{\pi \rho_w N \Gamma(\mu+4)}{6\rho q_c \Gamma(\mu+1)}\right]^{1/3}$$

with  $\rho$  air density,  $\rho_w$  bulk density of liquid water,  $q_c$  mass fraction of liquid water, N number of drops per volume.

(5) requires some explanations as for the inherent assumptions to be reproduced by the reader. A familiar model for the optical thickness (see e.g., Salby: Atmospheric Physics. Academic Press, 1996, Eq.(9.45)) gives

$$\delta = \frac{3}{2} \frac{\rho q_c dz}{\rho_w r_e}$$

which differs by a factor of 2 from (5). Please clarify.

• The nucleation rate (7) is connected with supersaturation S. Small but inevitable errors in vapour concentration  $q_v$  signify huge relative deviations in S. Can you estimate the resulting uncertainty in the nucleation rate?

Do you have a full prognostic equation for supersaturation S or do you use saturation adjustment to calculate S? In the second case, some more information is required for the calculation of the nucleation rate by (7). How do you get a supersaturation S > 0despite adjustment?

The uncertainty of calculation of S occurs in all schemes using an equation such as (7).

I wonder wether it is helpful to introduce more physical details on the nucleation rate as long as the basic property S carries such an uncertainty. Please comment.

The size of a freshly nucleated droplet is to be prescribed. What do you assume?

• Problem of avaraging.

p. 7, Figs. 4,5. Cloud water path is a property defined for the whole air column. Cloud effective radius, cloud droplet number concentration, and sulfate aerosol number concentration are defined locally, and for a grid point model the data are interpreted to be representative for the grid cell. For which level are the given data relevant? If they are vertical averages, please discuss, how the vertical average is calculated, how cloud free layers are considered, how the result is to be interpreted, etc. This point is even more complicated for the local variable  $r_e$ , which depends nonlinearly on the local variables N and  $q_c$ . Likewise, optical thickness is defined for a certain layer of thickness dz, maybe the layer where the respective  $r_e$  holds. The presented fields depend on the averaging method.

The same question arises for the daily averaging procedure and concerns also liquid water path. It concerns both, model and satellite data.

Please explain, and correct the discussion where neccessary. See Specific Points.

- Drop number concentration, liquid water content and path, optical thickness, and effective radius are interrelated, not independent of each other. Fig. 4 shows a strong correlation between optical thickness and cloud water path, as expected. The effective radius distribution shows a different pattern, somewhat inversely to the drop number concentration in Fig. 5; for same liquid water content, a lower  $N_d$  means a larger  $r_e$ , see e.g. the relationships (3) (5). This relation should be taken into account in the interpretation of Figs. 4 and 5. For the discussion of the improvement of COSMO-MUSCAT to COSMO-2M it would be helpful to include the COSMO-2M-fields in Fig. 4 besides (or instead of) the difference fields.
- The choice of the parameters  $C_{ccn}$  (p. 4 bottom) is a good general guess, however, not a universal constant. Did you do a similar run with modified  $C_{ccn}$ -values to check the influence - in opposition to the influence of the full interactive treatment with MUSCAT? COSMO-MUSCAT seems to result in smuch smoother distributions than COSMO-2M, in particular Fig. 5. Do you have an explanation?
- The aerosol-cloud-radiation interaction is an important point, since it affects directly the energy budget. Unfortunately, the discussion is limited to a description of Figures 6, 7, and no information on the cloud related parameters of COSMO-2MR are given. Either this aspect should be strengthened or skipped.
- The wording and the comparison can be more straightforward and more precise throughout the paper. Please work over the whole text. This concerns in particular the data intercomparison, which is done on a subjective basis phrasing like 'the differences are small'. Please quantify your statements for objective conclusions. Otherwise, e.g., the conclusion of superiority of COSMO-MUSCAT is not a priori clear from the case study, in particular since the difference between the MODIS data and each model result is larger than the difference between two model versions.

Please also interpret systematic differences in terms of the model modifications.

Might it be possible that parts of the differences between data from simulation and satellite are due to a) different cloud distributions and b) different instants of time used for the daily average?

**Specific points**

- Introduction: The section can be written in a more compact way. In particular, the 1and 2-moment schemes should be discussed primarily with regard to the aerosol-clooud and cloud radiation feedbacks.
  - 1.33: What is the outcome of Seifert et al. (2012)?
- p. 5, subsubsection 2.1.1. should read 2.2.

This short para has the character of an introductory explanation, but none of the methods is explained. Please give some more informations, e.g. in form of a short table as overview of all satellite data sources (ISCCP, CALYPSO, CERES, MODIS ...?), including informations of spatial and temporal resolution for the averaging aspect. Do you use all mentioned satellite data?

l. 19: I do not understand 'the assumptions for the satellite retrievals' in this context.

COSP is important for the paper. Please explain what the simulator does, at least the input and output data, and what kind of errors may occur.

What kind of spatial and temporal averaging is done? E.g., how many output times do you have for COSP- and for satellite data to determine a daily mean value? Can the averaging procedures produce a bias in the results, maybe the difference in daily averaged cloudiness in Figure 3?

What is the physical interpretation of a 'daily mean cloud cover'? 12h cloud free plus 12 h full cloud cover results in 50% cloudiness?

- p.5 Section 3.1: The synoptic situation should be described for the situation on 17 February, the day of the later discussion and evaluation.
- p.6 l. 16. 'Northerly wind'? Fig. 2 shows mostly south-westerly winds over the Atlantic.
- p.6 l. 18-19. Please precise the sentence 'The cold continental air mass ...'.
- In Section 3, you use 3 version of the COSMO model and several satellite data sets for mutual intercomparison. Please make clear everywhere, which respective data sets are compared, and break the passages of different intercomparisons. Please use always the same expressions. E.g. p.7 l. 2. Which two model versions? What is the 'MODIS simulator' (also l. 30)?
- Section 3.2 (in particular) contains inconsistencies in wording and notation compared to the rest. E.g., optical depth  $\delta$  vs.  $\tau_c$ , COSP satellite simulator vs. MODIS satellite simulation? Please unify.
- p.7 l.11pp. The spatial structure of the fields are similar. On the linear scale, I would not agree to 'slightly larger' (l. 11) or 'slight underestimation' (l. 18). I am well aware that both data sets are subject to many sources of error, hence a similar field structure and a similar order of magnitude should be acceptable, but not whitewashed.

p. 7 l. 14pp. The strongest differences do not occur near the Atlantic coast, but in the most western part of the domain. I have the impression that the model does not catch these clouds. Please clarify.

- p. 7 l. 19. Correct unit of cloud water path.
- p.7 l.20pp Fig.4 g-i. I do not follow your interpretation. The differences should be seen in relation to the signal. The least (relative) difference should be seen in the LWP, since the amount of condensate is primarily determined by other than microphysical processes and is to be seen in relation to the change in cloud ice. The sequel of e.g., red and blue

bands over the Biscaya may be a phase shift. A decrease of  $r_e$  by  $10\mu$ m is of the order of the signal, not a 'slight reduction'.

Please precise. I agree with your conclusion of l. 27-28. However, I cannot see the superiority of COSMO-MUSCAT from the presente material.

- p.7 l. 25. Again: Not 'slight' and 'little'.
- p.8 l.3. 'cloud microphysics are modified'. If this is worth mentioning, then please be more precise.
- p.8 l.3. Please explain what you mean by 'better agreement'. Allgemeine FRAGE!!
- p.8 l.6. Fixed CCN= 300cm-3 in COSMO-2M? This is in contradiction to Section 2.1, telling Nccn is given as function of S.
  l. 32. Similar: 'constant cloud condensation nuclei profile'?? Please clarify.
- p.8 l. 9pp. The aerosol NUMBER (not 'mass') concentration is given in Fig. 5c. Could you please comment on the fact, that  $N_{sulfate}$  is so much larger than  $N_d$  for COSMO-MUSCAT? Is the result of Boucher and Lohmann (1995) transferrable to your model concept?
- p.8 l.14pp Please revise the para.
  'the model exhibits more clear grid points.' What do you mean?
  'The model is unable to capture sub grid scale cloud patterns': A subgrid scale cloud cannot be captured by the microphysics parameterization of Seifert and Beheng (2012) or similar ones. You would need a different tool.
  'the satellite may overestimate the retrievals.' What do you mean?
- p. 8, Section 3.3. l. 25. '(20 to 20 W m-2'??' 'some regions': Please precise.
  Fig. 6. The colorbars are differently scaled for most of the subfigures. Sometimes this is straightforeward (e.g., a and f vs. b and f), sometimes, however, confusing (e.g. a vs. c, j vs. l). Please unify the scaling.

Please also consider to plot the net UPWARD LWF to have the colors consistent to the SWFs, e.g., blue for weak differences. Same for Fig. 7.

Fig. 7 a-d contains is repetition of Fig. 6 e-h. Use the difference fields COSMO2M rad minus CERES instead.

1. 27/28. I cannot follow the statement 'the differences are neither systematic nor large'. Please interpret the radiative flux differences also in terms of the cloud properties.

• p. 9 l. 7pp

Please check the conclusions with regard to the above points for more precise statements. Conclusion 1. If you refer to the model runs COSMO-2M and COSMO-MUSCAT, please say so. Then, this statement does not agree with p.7 l. 20-29. Please clarify. Conclusion 2. Precise the 'modified model simulation'.

- p9 l21. Missing reference.
- If a paper is written by two authors, please cite as 'A and B (1999)'.
- p.11: Citation of IPCC is incomplete.
- Please check ALL figures w.r.t. wording within the plots and in the legends. E.g., in Fig. 2 'Temeperature', in Fig.3 'MUCAT', in Fig. 4 g-i 'CSOMO2M'.

---

## Author Comment (AC1) · 15 Dec 2016

Model version number (COSMO-MUSCAT(5.0)) is included in the revised manuscript

---

## Author Comment (AC2) · 15 Dec 2016

[12pt]article geometry a4paper, total=170mm,257mm, left=25mm, top=20mm,

[utf8]inputenc graphicx  sfmath [hyphens]url  graphicx

**Reviewer Comments:1**

**Major Revisions:**
*The authors present two extensions to the regional atmosphere aerosol chemistry*

*model COSMO-MUSCAT. As a first extension, the authors now use the two-moment scheme from COSMO (Seifert and Beheng). The cloud condensation nuclei (CCN) information needed by this scheme comes from MUSCAT (instead of constant, prescribed CCN profiles), following Boucher and Lohmann 1995 and taking sulfate mass as a CCN proxy. The second extension concerns the cloud optical depth in the radiation scheme, which now accounts for droplet-size, via the cloud effective radius, following Martin et al. 1994.*

*To evaluate the effect of the two new features on code performance, the authors consider a ten day test period (February 15 to 25, 2007). Of this period they focus, however, mostly on one single day (February 17). The simulations are run in forecast mode, thus are not nudged. Variables considered for the evaluation are different cloud related quantities (cover, optical thickness, effective radius, water path, droplet number concentration) and shortwave and longwave net radiation at the surface and the top of atmosphere (TOA). Comparison is done among different model versions (COSMO2M, COSMO2M.rad, COSMO-MUSCAT) and with satellite data (CERES, MODIS, ISCCP) and typically comes in the form of maps.*

*The authors find significantly improved performance of the new code version when comparing modeled and satellite based cloud effective radius and cloud droplet number concentration. Improvements are less pronounced for other quantities like cloud optical depth, cloud water content, or cloud fraction.*

*The topic - effect of more elaborated aerosol-cloud treatment in a regional climate model - is of interest. A number of corresponding models exists (e.g. WRF or also COSMO, Zubler et al. 2011), yet given the complexity of the topic a larger number of models whose results can then be compared are clearly desirable. The study thus is of interest.*

*The study fits the scope of GMD, but requires major revisions to meet GMD standards.*

**Major Points**

*1), Precision and / or clarity of statements could generally be improved. Two examples*

GMDD
*in the following, more can be found under 'minor points' below.*

*Evaluation is currently done essentially via comparing maps and arguing that things look similar or that there is a slight increase, minor change, a largest change etc. What are the numbers behind such statements? Only few are given. Regional averages, variability, correlations, scatter plots etc. would allow for better quantitative comparison of the different models among themselves and with the satellite data.*

*What is the resolution (space and time) of the satellite data you use for evaluation? Are model averages based on data from each model time step or based on output data? If output: hourly or less frequent?*

ANS: In the revised manuscript a quantitative analysis is included with statistical representation of cloud microphysical properties as probability density functions (PDFs) corresponding to model (COSMO-MUSCAT) and satellite, which can account for different resolution of model and satellite observations (manuscript Figure 5).

In this study we have used MODIS, CERES and ISCCP satellite products for comparisons. The CERES data sets are daily products with spatial resolution of $1°$, the given overpass is compared with modeled daily average value. For the case study, ISCCP daily product with spatial resolution $2.5°$ are used. Further, MODIS level-2 swath data for the time 8.00 to 14.00 UTC are aggregated to model domain (overpass times over the domain). For comparison MODIS products are re-gridded to model domain.

*2) You state that you have run the model for 10 days in forecast mode, February 15 to 25, 2007. But you use only one day (February 17) for model evaluation. Why? More importantly, is a one day forecast enough to evaluate the different models? The model may still, above all, be in an adapting stage after only one day (see e.g. Cossu Hocke, GMD, 2014). This may also apply to aerosols, a key element in your study, but with lifetimes on the order of days. While the forecast mode of your simulations makes comparisons more difficult as time evolves, you may still check whether, for example, CCN and cloud optical thickness evolve in concert over the ten days of your simulation. Please comment on possibilities and limitations of your one day forecast comparison.*

*Or re-run your simulations in nudged mode an compare them over a longer time.*
ANS: The main objectives are to replace the constant aerosol number concentration in COSMO two-moment scheme with, gridded aerosol information from MUSCAT model and incorporate COSP satellite simulator in the model. For model evaluation we have selected 17 Feb 2007, which is the 3rd day of the simulation, it may reduce the uncertainty in the model prediction as the time progress, also it would get sufficient time for MUSCAT model to evolve the transport processes. In further response to reviewer's remark, additional days were analyzed and compared with satellite observation and they are also in agreement (Figure 1 & 2).

**Minor Points:**
*p.2, l.30-35: As you point out that other groups have already coupled COSMO with two-moment cloud microphysics to an aerosol module, including droplet-size aware radiation. Please explain to the reader how your work differs from these existing, closely related approaches.*
ANS: Seifert et al., 2012 has included the cloud-radiation coupling in which effective radii of ice particles and cloud droplets are calculated in the microphysics scheme and passed to the radiation scheme, which is not available with official version of COSMO with 2-moment and has some issues (A. Seifert, personal communication). COSMO-ART is also implemented with double moment scheme, which uses droplet activation based on Bangert et al., (2001), instead of activation rate equation (17) of Seifert et al., (2006).

*Section 2.1 I find rather difficult to read as information on different codes (COSMO, COSMO-MUSCAT, and MUSCAT) as well as different code versions (current version, several other versions) is tightly interleaved. It is not always obvious whether what is stated applies to COSMO, COSMO-MUSCAT, the former or present code etc. For example, p.4, l.23ff: to which model do these equations now apply? COSMO2M?*

*If so, how do the CCN numbers given here (1.26e9 and 1.0e8) go together with the
300 mentioned p.8,l.6? I further guess, but this is not really clear from the text, that
equation 7 also applies to COSMO-MUSCAT, but with Cccn taken from MUSCAT and
probably k and Smax the same as in COSMO. Please clarify.*
ANS: p4, l.23ff: These equations are applied to COSMO model with two-moment,
which is revised in the manuscript (COSMO2R). The CCN numbers are applied for
continental and maritime conditions ($1.26\times10^9$ and $1.0\times10^8$), however in this study
we have used intermediate aerosol which has a CCN value of $3.0\times10^8$, which is also
revised in the manuscript.

*p.4, eq.3: What is Gamma?*
ANS: It is gamma diustribution function.

*p.5,l.3: 'aerosol mass concentration information from the MUSCAT model'. Where
does MUSCAT get that information from? Are aerosol emissions prescribed? If so,
where from? Or concentrations? What are these emissions / concentrations?*
ANS: The emission inventory in MUSCAT model is proved by TNO for the Air Quality
Model Evaluation International Initiative (AQMEII) project (Pouliot et al., 2012).

*In section 3, evaluation against satellite products: is snow cover an issue?*
ANS: In satellite retrievals, mainly thin clouds are affected by snow cover, which could
be rather ignored.

*p.6, l.6: 'with radiation coupled with microphysics': you mean the cloud effective radius
following Martin et al. 1994 (your equations 3 to 5) is used? But for a fixed CCN
number of 300?*
ANS: p.6, l.6: COSMO two-moment with radiation coupled to microphysics (COSMO-
2MR), with fixed CCN ($3.0 \times 10^8 m^{-3}$), which uses equation 3 to 5 in radiation scheme.

[Figure]

*p.6,l.27: Unless you have further evidence that for the concrete case ISCCP indeed underestimates cloud cover over the Atlantic, a fairer formulation may be that besides the model having a problem it could also be that the satellite has a problem.*
ANS: The problem can arise from both side.

*p.7, l.2: 'In the two model versions...' You consider three model versions, COSMO2M, COSMO2M.rad, and COSMO-MUSCAT. While by and by one finds out what two versions you mean here, please state so explicitly.*
ANS: Revised as suggested

*p.7, l.3ff: Can you comment further on this screening for the liquid phase in MODIS and the models? How dominant is the liquid phase in either one?*
ANS: In MODIS satellite retrievals, liquid clouds are screened by setting "Cloud_Phase_Optical_Properties flag=2" and the COSP satellite simulator is able to simulate cloud optical properties for liquid and ice phases separately.

*p.7, l.9: 'In both cases it varies between 2 and 50...' Does the real quantity vary in that range or just your colormap? Also, figure 4a shows clearly much more red color than figure 4d. Reducing the comparison of the two panels to their range skips this aspect. In that sense, the patterns are not similar, as claimed in the text. Also, in wide parts where there is substantial cloud optical depth, the satellite based value is about twice as large as the model value. I would not call this a slight difference but a factor of two. This is another example where precision could be improved (major point 1).*
ANS: For liquid clouds, we have considered the lower limit of cloud optical depth is 5, which is based on the study by Sourdeval et al., (2015), and the maximum value is 54.0, which are revised in the manuscript. This part is also revised as suggested.

*p.7, l.14 ; High values for cloud effective radius are also seen over land. And, as above (p.6, l.27), it is not obvious that the flaw is with the satellite data.*
ANS: p.7, l.14 has been removed from the manuscript, and p.6, l.27.: This may either due to the coarse (280 km resolution) resolution of the satellite observation or poor parameterization of clouds in the model.

*p.7, l.21: What do you mean by 'largest impact'? Largest in what sense? Change in mean? Median? Percent or absolute? Per grid box?*
ANS: p.7 l.21 has been removed from the manuscript.

*p.7, l.25: 'slight increase': what is slight?*
ANS: p.7, l.25: In the case of cloud optical depth and cloud water path, both generally show an increase despite of reduction in a few areas.

*p.7, l.26: 'The cloud optical depth shows a variation in the range of +/-20...' Variation over what? Spatially? Within a model? Please clarify.*
ANS: The revised parameterization in coupled model has made modification in spatial distribution of cloud optical depth in the range of $\pm$ 15 and the liquid water exhibits a variation in the range of $\pm$ 0.12 $kgm^{-2}$.

*p.8, l.8: There is a wide region (red in figure 5d) with CCN of at least 300, i.e., the prescribed CCN value in COSMO2M. Yet the CDNC in COSMO-MUSCAT is much lower than in COSMO2M also in this region. Why?*
ANS: The sulfate aerosol number concentration in Figure 6d is vertically averaged. The high CCN values are mainly occur below the cloud layers because the high pressure in this region results in trapping CCN below boundary layer.

*p.8, l.14: 'While comparing with high resolution MODIS satellite products...': These have not yet been introduced, I think.*
ANS: Rephrased to MODIS level-2 products

*Section 3.3.: It would be interesting to elaborate a bit more on radiation. For example, the differences between COMSMO2M and COSMO2M.rad seem to be larger over sea than over land. True? Do the large differences (more downward SW and upward LW at the surface especially over sea) go hand in hand with reduced cloud optical thickness? Change in cloud effective radius? Cloud cover? Regarding the comparison with CERES: what area means of CERES and models? Given that you look at February (little radiation, snow cover, short days) and a cloud cover close to 100 percent over wide regions: how reliable are CERES surface fluxes?*
ANS: The effect of aerosol-cloud-radiation interaction can be seen to larger extend over ocean than over land, especially for surface net downward short wave and long wave fluxes. The cloud microphysics modification results more surface SW and LW radiation over sea. During winter the uncertainty in the CERES flux is little higher due to large zenith angle (Guo et al. 2007).

*p.8, l.30: 'This paper presents an initial approach to the modification of Seifert and Beheng (2006) two-moment scheme in the COSMO model.' This is not true. Other groups have done this before, e.g. Zubler et al. (2011) whom you cite.*
ANS: Rephrased to : This paper discusses the modification of Seifert and Beheng (2006) two-moment scheme in the COSMO model.

*p.9,l.2: Maybe state that this parameterization takes sulfate mass as a proxy, it is not a full grown aerosol module like SALSA, M7, etc.*

ANS: Revised as suggested

*p.9, l.8: 'In terms of the cloud distributions, this modification has only a minor effect.' Given that you compare the second day of forecast simulations in winter, this is not truly surprising. To what degree is this finding just due to large scale weather condition or your initialization?*
ANS: We do not have another simulation to clarify.

*p.9, l.9: What means daily averaged when you consider only one day in the first place?*
ANS: In the case study, model results are compared with satellite products. While comparing with ISCCP, the model data is daily averaged because the satellite products are daily products. Whereas in MODIS (Terra) level-2, overpass observations are considered, which is between 8.00 - 14.00 UTC over the domain, so the models outputs are averaged between 8.00 to 14.00 UTC.

*p.9, l.10: 'The modified model simulations are in broad agreement with satellite observations.' I would argue that all your simulations are in 'broad agreement'. However, you see some improvements (as you state) in your modified version.*
ANS: The interactive treatment of aeorosls in COSMO-MUSCAT simulations show an improvement in the cloud microphysical properties. Further, the PDF analsysis has contributed to a quantitative comparison of model reuslts with satellite observations.

*p.9, l.15: '...only minor changes in terms of the radiation budget were found.' Looking at figure 6, i-l, I would not call these changes minor. In wide regions they are on the order of a factor of two.*
ANS: Again, considerable changes ($\approx$ factor of 2) in terms of the radiation budget were also found. The new approach now, however, allows to explicitly take into account the

radiative effects of aerosol-cloud interactions.

*Figure 1: Why does it say "M7 to be implemented"? In the reference you cite, Wolke et al. 2012, it is stated that M7 is implemented. Please explain. And if M7 does indeed not form part of your model version, remove it from Figure 1.*
ANS: In this version of COSMO-MUSCAT(V5.0), M7 is not yet implemented. Also Figure 1 is modified without M7.

*Figure 2: What is the data source?*
ANS: Initial state of model simulation.

*Figure 3: Does cloud cover from COSMO2M, and possibly COSMO2M.rad, look similar to cloud cover from COSMO-MUSCAT?*
ANS: The COSMO2M and COSMO2R cloud cover looks similar, however COSMO-MUSCAT cloud cover has been modified due cloud microphysics modification.

*Figure 4: While the figure is useful, some more quantitative comparisons would also be useful, e.g. area means, variability, scatter plots... For example, what is the are mean cloud water path in 4i? And how does it compare wit the area mean of 4f? Figure 4i looks as if there is above all a change in spatial distribution of the cloud water path, not of total cloud water path (area mean). The same question may be asked for figures 4g and 4h.*
ANS: It would be difficult to have a correlation of area mean with satellite observation, because of the different grid points and the satellite products are combined for different swaths. Also the area mean would increase the uncertainty. To overcome this we have compared PDFs of cloud microphysical properties.

*Figure 4: How different would the figure be if you were to compare COSMO-MUSCAT with COSMO2M.rad? Put differently, are the differences mainly due to the variable CCN or also to the size-aware radiation?*

ANS: There are some difference, if we compare COSMO-MUSCAT with COSMO2MR, however, it can be more clear if we compare COSMO2MR with COSMO2M. It is noticed that the major difference is due to CCN (Figure 3).

*Figure 5c: Point out that this is not a MODIS product but a derived quantity. Also, why is there hardly anywhere a CDNC greater than 10? After all, there is cloud cover all over the place and CDNC=10 or smaller is very low.*

ANS: CDNC can be derived from MODIS cloud optical depth $\tau_c$ and effective radius $r_e$ (Quaas et al., 2009), which is given by,

$$N_d = \alpha \tau_c^{0.5} r_e^{-2.5} \tag{1}$$

where $\alpha$ = $1.37 \times 10^{-5} m^{-0.5}$. In the above equation the lower limit of $\tau_c$ and $r_e$ are constrained to 5 and 2. This would result in removing lower CDNC values.

*Figure 6: On the western boundary of the domain, there seems to be a boundary effect. Can you comment?*

ANS: The boundary effect in the difference can be ignored, which arises due to different physics in COSMO and GME.

*Figure 6 j-l: Given that the color scale is saturated in wide regions in these plots, why not take it larger? Possibly even -40 to +40, as in 6i?*

ANS: Figure 6 and 7 are revised

*Figure 7, a-d: Same figures as in figure 6 e-h. No need for duplication. You may consider replacing these panels with corresponding ones from COSMO-MUSCAT, so*

*one has all three models and CERES shown.*
ANS: Figure 7 is revised with COSMO-MUSCAT.

*Figure 7 6: What time is shown? February 17? 24h mean? Color scale in the SFC
SW plots could be reduced to maybe 200 instead of 260 to fully exploit the range of
the color table.*
ANS: Figure 7 is revised. It is for February 17, 24h mean.

*It maybe worthwhile to point out somewhere that you only changed the model but did
not (yet) re-tune it, e.g. to get reasonable 2m temperature or precipitation. You show
that the different codes give, for example, different cloud optical properties. But this
does not imply an overall better model performance.*
ANS: Although the two-moment cloud microphysics scheme in COSMO model has
been modified, the model did not re-tuned to get reasonable 2m temperature or
precipitation. This sentence is included in conclusion section.

*The language needs brushing, there are a number of sentences that do not work on
the language level. I give only two examples.*
ANS: The language has been revised.

*p.2, l.6/7: "Although regional models do not describe part of the large scale feedbacks
which are included in GCMs, regional modeling allowing for an optimal compromise.*
ANS: P.2, l.6/7: Even though regional models do not describe part of the large scale
feedbacks, it provides optimal compromise.

*p.8, l.8/9: "From figure 5c, the maximum aerosol mass concentration observed over
south eastern Europe, on the contrary Nd shows less.*

ANS: p.8, l.8/9: From figure 6d, maximum aerosol mass concentration is observed over south eastern Europe. On the contrary, $N_d$ shows less over the same region.

Fig. 1. MODIS Level-2 (a) cloud optical depth, (b) cloud effective radius, (c) cloud water path, COSMO-MUSCAT derived (day time averaged) (d) cloud optical depth, (e) cloud effective radius, (f) cloud water path for 23 February 2007.

Fig. 2. MODIS Level-2 (a) cloud optical depth, (b) cloud effective radius, (c) cloud water path, COSMO-MUSCAT derived (day time averaged) (d) cloud optical depth, (e) cloud effective radius, (f) cloud water path for 24 February 2007.

Fig. 3. Comparison between COSMO2MR and COSMO2M for 17 February 2007.

---

## Author Comment (AC3) · 15 Dec 2016

**Reviewer Comments:2**

**Major Revisions:**

•*Whilst the paper gives a detailed introduction of the general benefits of regional models with online coupled aerosol-cloud, aerosol-radiation and aerosol-cloud-radiation interactions, such as WRF-CHEM (Grell et al. 2005), COSMO-ART (Bangert*

et al. 2011), COSMO-M7 (Zubler et al. 2011) and COSMO-ART/M7 (chapter 2 of: http://e-collection.library.ethz.ch/eserv/eth:48845/eth-48845-02.pdf), I would like to be given further motivation regarding the advantages of their particular approach and have it contrasted to existing approaches.

*My questions raised here, could guide such a discussion:*
1) *Why did you use the Boucher and Lohmann's (1995) empirical relation, rather than implementing newer approaches using Koehler theory that have previously been applied in COSMO-ART/COSMO-M7 such as Abdul-Razzak & Gan (2000), or Nenes & Seinfeld (2003)? What is your justification for only considering sulfate and ignoring nitrate contributions to CCN?*

ANS: Our objective is to replace constant $C_{ccn}$ in COSMO model with $C_{ccn}$ from MUSCAT model. This particular version of COSMO-MUSCAT (Version 5.0) is only available with aerosol mass concentration for different aerosol species. As an initial stage of two-moment scheme modification, we have used Boucher and Lohmann parameterization to convert sulfate aerosol mass concentration to $C_{ccn}$ number concentration. In the next step, M7 will be implemented and we could directly use aerosol number concentration in the two-moment scheme, along with new approaches.

2) *Whilst the relationship between cloud optical depth, effective radius and vertically integrated liquid water content is a commonly used diagnostic, it is wavelength independent. How did you deal with this issue when implementing this approach in a radiation scheme that computes the radiative transfer within 3 SW and 5 LW bands covering the entire wavelength spectrum?*

ANS: Even though the cloud optical properties are wavelength independent, the extinction coefficients are calculated from wavelength dependents variables. The

extinction, scattering and asymmetry parameters are depends on entire wavelength spectra (Zubler et al., (2011)).

• *Furthermore, a short motivation should be included regarding the chosen case study. Are all the clouds shown boundary layer stratocumulus?*

ANS: A short motivation is included in the manuscript. All clouds are not boundary layer stratocumulus, it consists of convective clouds too.

• *The model evaluation is performed on a single day. The authors argue that the forecast skill decreases with increasing lead time. Could you obtain better agreement and therefore obtain a longer evaluation period if you performed nudged simulations? Otherwise, a brief justification should be given that 1 day is a sufficiently long time period for your evaluation.*

ANS: Even though the model comparison via spatial distribution is performed only for a single day, more quantitative analysis (PDF comparison for single day and for entire simulation) has been carried out and included in the manuscript (Figure 5). The nudged simulation may improve the evaluation, however in this first stage we are mainly focusing on COSMO-MUSCAT coupling and COSP satellite simulator incorporation and its comparison with observations. A brief justification is included in the manuscript.

**General Comments:**

• *This paper talks about the coupling of aerosol-cloud interactions and aerosol-cloud-radiation interactions. However, it should be stated clearly (already in the abstract)*

*that these interactions are only included explicitly for warm-phase cloud processes.*
ANS: These interactions are applied for liquid water clouds due to complexity of the aerosol cloud interaction in ice clouds, which is mentioned in the manuscript.

• *The paper often talks about changes to the COSMO model in general. However, the modifications are applied not to the COSMO model in general, but to the COSMO version of COSMO-MUSCAT. Whilst I agree with the authors that CCN variability and aerosol-cloud-radiation coupling is not provided in the officially released code versions, such developments have already been included in other COSMO versions (COSMO-ART, COSMO-M7, COSMO-ART/M7). They are therefore not new to the COSMO code itself.*
ANS: Although, different COSMO versions (COSMO-ART, COSMO-M7, COSMO-ART/M7) have been modified, these models used different cloud microphysics (mainly droplet activation), for example in COSMO-ART the droplet activation based on Bangert et al., (2001), which replaces the activation rate of equation (17) of Seifert et al., (2006) and in COSMO-M7, it is treated with following parameterizations Leaitch et al. (1996), Lin and Leaitch (1997), and Lohmann (2002).

• *The paper is written in good scientific English. However, some sentences require re-writing (some of which are listed below). Furthermore, articles are missing in a few places.*

**Specific Comments:**

• *P2L6: Last sentence needs rewriting.*
ANS: Revised as suggested

- *P2L15: Sentence needs rewriting:"This approach, however,..."*
ANS: Revised as suggested

- *P2L31-32: Delete last sentence of this paragraph. It is mis-leading as all of the mentioned aerosol-coupled regional models include a coupling of the cloud microphysics to the radiation.*
ANS: Revised as suggested

- *P3L22:Sentence needs rewriting:"In this study, the COSMO model..." Equation 5: Change $\frac{1.5}{2}$ to $\frac{3}{4}$.*
ANS: In this study, COSMO model has been configured in a non convection permitting mode with uniform horizontal grid resolution of $0.25°$ ($\approx$28 km).

$$\tau_c = \frac{3\rho q_c dz}{2\rho_w r_e} \tag{1}$$

- *P4L23: Sentence needs rewriting: "IN the COSMO model, the aerosol..."*
ANS: Revised as suggested

- *P5L1-11: This discussion needs some clarification: How is the Smax issue raised overcome? You raise the issue of ignoring updraft velocity in the Boucher and Lohmann formulation, however, is this then not introduced by dS/dz\*w in Eq.7 (which admittedly is a very simplistic formulation of this relationship).*
ANS: In this case simulations are carried out for intermediate aerosol condition, in which $S_{max}$ is set at 2.0%. Boucher and Lohmann parameterization is bound to updraft velocity uncertainty, however it can be overcome by dS/dz\*w in Eq.7. This part is revised.

- *P5L20ff: Here you list all satellite products used for evaluation. Also include CERES*

*here, as you use it later.*
ANS: Revised as suggested

• *P5L27: I suggest to delete: In the upcoming... You have 1 ISCCP figure, 2 Modis
and 1 CERES. That is not massively unbalanced.*
ANS: Revised as suggested

• *P5L30: I suggest to delete last sentence of this paragraph.*
ANS: Revised as suggested

• *P5L25-30: Please rephrase. The point regarding satellite biases could be formulated
like: "One should keep in mind that the satellite products, just like models, are prone
to biases. Comparisons of satellite retrievals with ... have shown that.... Nonetheless,
spatial correlation of the cloud structures are well represented."*
ANS: Revised as suggested

• *P6L2: Sentence needs rewriting: "As the forecast time..."*
ANS: Revised as suggested

• *P6L10: Results are only shown for 17th. Not 15th – 25th. Please include description
of the cloud types of this domain (altitude, phase, average thickness, surface precip-
itation). When did you start your simulation, cause the 2nd day would be the 16th if
the simulation were started in the morning of the 15th? Also, why are is the 16th not
included in the analysis?*
ANS: In the revised manuscript, synoptic conditions for 15 Feb (Figure S1) and 17 Feb
(Figure 2) are included along with cloud type. Simulation started at 15 Feb, 00.00 hrs,
then 17-Feb would be the third day of the simulation (Revised in the manuscript).

• *P6L22-25: First 3 sentences should go into section 2.1.1 (methods).*
ANS: Revised as suggested

• *P7L1-4: First 3 sentences should go into section 2.1.1 (methods).*
ANS: Revised as suggested

• *P7L6: "The top panel shows...." This should be in Fig. Caption.*
ANS: Revised as suggested

• *P7L19: Please check units.*
ANS: The unit has been corrected to $kgm^{-2}$

• *P7L25-30: In this discussion quantitative statements should be included. For instance, the area mean changes + variability could be determined. It would help determine the signal from the noise in figures g to i.*
ANS: For quantitative comparison, PDF distribution of cloud products are included in the analysis.

• *P8L14: First sentence of paragraph need rewriting/clarification.*
ANS: Sentence rephrased: While comparing modified two-moment scheme results with MODIS level-2 satellite products, the model shows more cloud free (clear) grid points. This indicates that model is unable to capture the sub grid scale cloud patterns accurately (Jason and Thomas, 2008), which may be due to the coarse resolution (0.25°) of the model.

• *P8L19: Maybe rephrase title, because aerosol-cloud radiation interactions were already discussed in previous section (cloud optical thickness). Suggestion: "Impacts*

*on radiative balance" In general this discussion is not very precise. It would help the discussion if you relate observed changes in e.g. the SW fluxes to the decrease in cloud optical thickness...*
ANS: Revised as suggested

• *P8L30: Please rephrase. It is an initial approach to modify COSMO-MUSCAT only.*
ANS: Revised as suggested

• *P9L7: Finding 1 should be removed as it is not discussed in the paper.*
ANS: Finding 1 has been changed to: "The modified two-moment scheme results have been compared with two-moment version of COSMO model. In terms of the cloud distributions, this modification has only a minor effect".

• *P9L13: Paragraph missing after point 2.*
ANS: Revised as suggested

• *P9L22: Last sentence needs rewriting and one reference is missing.*
ANS: Sentence rephrased: This can result in more precise cloud droplet activation parameterization, involving different aerosol species as CCN, and thus improving the cloud droplet number calculation of Lohmann et al., (2007).

• *P10ff: A few references need changing. Some paper titles use capital letters. IPCC reference is incomplete*
ANS: Revised as suggested

• *Fig1: More detail should be given in the caption. I think this figure could be*

*improved/clarified. Why are the "emissions/land use" and M7 stand-alone and not connected? The figure could be clearer regarding the structure of the code. Are RACM and MUSCAT separate modules? Will M7 not be embedded in MUSCAT? Is the output really completely separate?*

ANS: Figure 1 has been modified, COSMO-MUSCAT is an online coupled modeling system, which uses COSMO meteorological fields to drive the model. Emissions and land use data are in input for MUSCAT model and RACM is a module, which is included in MUSCAT. In this particular version of COSMO-MUSCAT(V5.0), M7 not yet embedded. Also, the outputs are separate.

• *Fig2: I personally would not include all "H" and "L" markes. I would simply mark the center of the dominant low pressure and high pressure systems.*

ANS: Revised as suggested

• *Fig3/Fig4: Looking at these figures optically, I would arrive at different cloud fractions for Fig3b and Fig4d. Areas where the total cloud fraction of the model is 100%, the diagnosed optical thickness is 0 or just very small? Does the COSP simulator include subgrid-scale cloud water? If not, is that justified for cloud optical thickness?*

ANS: ISCCP satellite observations are available with 280 km resolution, for the comparison COSP derived ISCCP output (28 km) are re-gridded to 280 km, which is stated in the manuscript. The cloud water simulated at sub-column within model grids account for subgrid‒scale variability.

• *Fig4: Maybe regional/domain means + a measure of variability could be given to highlight the results more quantitatively?*

ANS: For quantitative comparison, PDFs are used, instead of regional means, which may add uncertainty.

- *Fig5: What is going on with the MODIS cloud droplet number estimate? Why is the spatial pattern so different than in Fig4a-c. Are these clouds just too shallow to obtain a good number estimate?*

ANS: CDNC can be derived from MODIS cloud optical depth $\tau_c$ and effective radius $r_e$ (Quaas et al., 2009), which is given by,

$$N_d = \alpha \tau_c^{0.5} r_e^{-2.5} \tag{2}$$

where $\alpha$ = $1.37 \times 10^{-5} m^{-0.5}$. In the above equation the lower limit of $\tau_c$ and $r_e$ are constrained to 5 and 2. This would result in low CDNC in figure 6c.

- *Fig6/Fig7: I find the color scale depicting the fluxes very misleading. I would suggest using a pure blue color scale for LW fluxes and pure red for SW fluxes.*

ANS: Revised as suggested
* * *

---

## Author Comment (AC4) · 15 Dec 2016

**Reviewer Comments:3**

*The authors present a numerical study on aerosol, clouds and radiation with mutual interactions and compare the results with satellite derived data. . This is an important topic, since all cloud-related processes pose a severe problem in weather forecast and climate modeling. The paper contributes to the ongoing research by examining the effect of mutual interactions of these processes and the improvement of atmospheric*

*models. This is worth to be published. The presentation is concise, length and number of figures are appropriate. However, the presentation is partly very vague and not consistent throughout the paper. The differences between the resp. data fields are inspected by eye, but not quantified. Therefore, it is difficult to follow the conclusions. Errors in some equations may be typos. Yet, before publication, I suggest some substantial revisions. Please see the major points and specific remarks below.*

**Major Points:**

• *In mid-latitude winter I expect that the ice phase plays an important role in the development of clouds and precipitation (Bergeron-Findeisen effect!), and you use the Seifert and Beheng (2006) scheme for mixed phase clouds. The paper, however, is devoted to the liquid phase alone.*

*Please discuss the effect of the modified treatment of drop nucleation on the ice phase properties, since a modification in one path of condensate formation is connected with an opposing trend in other path(s).*

*How do you determine the effective radius under cloud free conditions?*

ANS: Even though SB scheme is for mixed phase clouds, heterogeneous ice nucleation is not included in the official version of COSMO. Also, Seifert et al., (2012) has demonstrated the importance of heterogeneous ice nucleation by adding Philip et al., (2008) parameterization (not available with official version). Hence, to reduce the uncertainty in aerosol cloud interaction, we have restricted our analysis to liquid phase clouds only. Moreover, our main objectives are to modify the fixed CCN in two-moment scheme with online coupled COSMO-MUSCAT model and incorporate COSP satellite simulator in it. In future we will be addressing aerosols as INP. Additionally, cloud ice optical properties on 17 February shows that (Figure 1), the study area is dominated by liquid phase clouds rather than ice clouds.

In satellite observation, cloud droplet number concentration can be derived from MODIS cloud optical depth $\tau_c$ and effective radius $r_e$ (Quaas et al., 2009), which is

given by,

$$N_d = \alpha \tau_c^{0.5} r_e^{-2.5} \qquad (1)$$

where $\alpha = 1.37 \times 10^{-5} m^{-0.5}$. In the above equation the lower limit of $\tau_c$ and $r_e$ are constrained to 5 and 2. This would result in low CDNC value across the domain (Figure 6c).

•*Equations (3) - (5) (3) holds if the cloud drop size distribution is used with the internal coordinate drop diameter D, not radius r. Then, (4) follows as*

$$\lambda = \left[ \frac{\pi \rho_w N \Gamma(\mu + 4)}{6 \rho q_c \Gamma(\mu + 1)} \right]^{\frac{1}{3}} \qquad (2)$$

with $\rho$ is air density, $\rho_w$ is bulk density of liquid water, $q_c$ is mass fraction of liquid water, N number of drops per volume.

*(5) requires some explanations as for the inherent assumptions to be reproduced by the reader. A familiar model for the optical thickness (see e.g., Salby: Atmospheric Physics. Academic Press, 1996, Eq.(9.45)) gives*

$$\delta = \frac{3 \rho q_c dz}{2 \rho_w r_e} \qquad (3)$$

*which differs by a factor of 2 from (5). Please clarify.*

ANS: The cloud droplet size distribution is represented by gamma function, which is used with drop diameter.

$$\phi(D) = N_o D^\mu e^{-\lambda D} \qquad (4)$$

where D is droplet diameter, $\lambda$ is slope parameter and $\mu$ is spectral shape parameter.
Whereas the effective radii for droplets and cloud ice are obtained directly by dividing the third and second moments of the size distribution given by (Morrison et al., 2008)

$$r_e = \frac{\Gamma(\mu + 4)}{2\lambda\Gamma(\mu + 3)} \tag{5}$$

Equation 5 is corrected to

$$\delta = \frac{3\rho q_c dz}{2\rho_w r_e} \tag{6}$$

• *The nucleation rate (7) is connected with supersaturation S. Small but inevitable errors in vapor concentration $q_v$ signify huge relative deviations in S. Can you estimate the resulting uncertainty in the nucleation rate?*
*Do you have a full prognostic equation for supersaturation S or do you use saturation adjustment to calculate S? In the second case, some more information is required for the calculation of the nucleation rate by (7). How do you get a supersaturation S > 0 despite adjustment?*
*The uncertainty of calculation of S occurs in all schemes using an equation such as (7).*
*I wonder wether it is helpful to introduce more physical details on the nucleation rate as long as the basic property S carries such an uncertainty. Please comment. The size of a freshly nucleated droplet is to be prescribed. What do you assume?*
ANS: In COSMO model, nucleation rate is parameterized as a function of grid scale supersaturation and vertical velocity. It uses saturation adjustment to calculate S. Also, It is logical to use nucleation scheme explicitly depending on supersaturation in combination with saturation adjustment, which is done by applying an operator splitting method (Seifert et al., 2006). For SB parameterization the arbitrarily chosen small droplet mass is given by $1 \times 10^{-12} kg$, and corresponding size of freshly nucleated drop is 6.2 $\mu m$. A detailed explanation is available in Seifert et al., 2006.

- *Problem of averaging.*

*p. 7, Figs. 4,5. Cloud water path is a property defined for the whole air column. Cloud effective radius, cloud droplet number concentration, and sulfate aerosol number concentration are defined locally, and for a grid point model the data are interpreted to be representative for the grid cell. For which level are the given data relevant? If they are vertical averages, please discuss, how the vertical average is calculated, how cloud free layers are considered, how the result is to be interpreted, etc. This point is even more complicated for the local variable re, which depends nonlinearly on the local variables N and qc. Likewise, optical thickness is defined for a certain layer of thickness dz, maybe the layer where the respective re holds. The presented fields depend on the averaging method.*

*The same question arises for the daily averaging procedure and concerns also liquid water path. It concerns both, model and satellite data. Please explain, and correct the discussion where necessary. See Specific Points.*

ANS: COSMO and COSMO-MUSCAT models are incorporated with COSP satellite simulator (Bodas-Salcedo et al., 2011). The variable such as, cloud water path, cloud optical depth, effective radius and sulfate aerosol number concentration are derived from COSP satellite simulator, which are vertically averaged.

To produce similar output to satellite data, COSP requires grid mean vertical profile of temperature, humidity, hydrometer mixing ratio, cloud optical thickness and emissivity, surface temperature and emissivity from the model. It produces the output comparable with satellite data in three steps. First it address the mismatch between model and satellite pixel, second vertical profiles of individual sub-columns are passed to each instruments and finally COSP statistic module gather the output from all instruments (Bodas-Salcedo et al., 2011). The above paragraph is included in the revised manuscript (Section 2.2).

- *Drop number concentration, liquid water content and path, optical thickness, and effective radius are interrelated, not independent of each other. Fig. 4 shows a strong*

*correlation between optical thickness and cloud water path, as expected. The effective radius distribution shows a different pattern, somewhat inversely to the drop number concentration in Fig. 5; for same liquid water content, a lower Nd means a larger re, see e.g. the relationships (3) - (5). This relation should be taken into account in the interpretation of Figs. 4 and 5. For the discussion of the improvement of COSMO-MUSCAT to COSMO-2M it would be helpful to include the COSMO-2M-fields in Fig. 4 besides (or instead of) the difference fields.*

ANS: There was an error in Figure 4, which is corrected in the manuscript. In the corrected version, there is correlation between cloud optical depth, effective radius and liquid water path. The difference between COSMO-MUSCAT and COSMO-2M would more relevant than COSMO-2M.

• *The choice of the parameters Cccn (p. 4 bottom) is a good general guess, however, not a universal constant. Did you do a similar run with modified Cccn-values to check the influence - in opposition to the influence of the full interactive treatment with MUSCAT? COSMO-MUSCAT seems to result in much smoother distributions than COSMO-2M, in particular Fig. 5. Do you have an explanation?*

ANS: In the interactive model (COSMO-MUSCAT) the general guess has been replaced by $C_{ccn}$ calculated using equation 8.

In the revised manuscript, Figure 6(a to c) has been modified. In order to compare the model simulation with satellite observations, we have used equation (9) to compute model $N_d$, as the COSP simulator can provide cloud optical depth and effective radius similar to MODIS satellite.

In COMSO-2M, we have used intermediate aerosol ($C_{ccn} = 3.0 \times 10^8 \ m^{-3}$), when it comes to COSMO-MUSCAT interactive simulation, it uses gridded $C_{ccn}$ information from MUSCAT.

From Figure 6d it is noticed that the maximum value of sulfate aerosol number concentration is in the order of $3.0 \times 10^8 \ m^{-3}$, however the droplet activation is controlled by several other meteorological properties such as vertical velocity and micro-physical

links.

• *The aerosol-cloud-radiation interaction is an important point, since it affects directly the energy budget. Unfortunately, the discussion is limited to a description of Figures 6, 7, and no information on the cloud related parameters of COSMO-2MR are given. Either this aspect should be strengthened or skipped.*
ANS: This part has been revised.

• *The wording and the comparison can be more straightforward and more precise throughout the paper. Please work over the whole text. This concerns in particular the data intercomparison, which is done on a subjective basis phrasing like 'the differences are small'. Please quantify your statements for objective conclusions. Otherwise, e.g., the conclusion of superiority of COSMO-MUSCAT is not a priori clear from the case study, in particular since the difference between the MODIS data and each model result is larger than the difference between two model versions.*
*Please also interpret systematic differences in terms of the model modifications. Might it be possible that parts of the differences between data from simulation and satellite are due to a) different cloud distributions and b) different instants of time used for the daily average?*
ANS: The wording and the comparisons are revised in the manuscript.

**Specific points**

• *Introduction: The section can be written in a more compact way. In particular, the 1 and 2-moment schemes should be discussed primarily with regard to the aerosol-cloud and cloud radiation feedbacks.*
*l.33: What is the outcome of Seifert et al. (2012)?*

ANS: Introduction is revised. l.33: Seifert et al. (2012) reported that in COSMO model, radiative aerosol induced effects are more relevant than effect on precipitation. They have shown that, one-moment scheme has a strong positive bias in maximum 2m temperature. This difference between one-moment and two-moment scheme may partly explained by different cloud-radiation coupling.

- *p. 5, subsubsection 2.1.1. should read 2.2. This short para has the character of an introductory explanation, but none of the methods is explained. Please give some more informations, e.g. in form of a short table as overview of all satellite data sources (ISCCP, CALYPSO, CERES, MODIS ...?), including informations of spatial and temporal resolution for the averaging aspect. Do you use all mentioned satellite data?*
*l. 19: I do not understand 'the assumptions for the satellite retrievals' in this context. COSP is important for the paper. Please explain what the simulator does, at least the input and output data, and what kind of errors may occur.*
*What kind of spatial and temporal averaging is done? E.g., how many output times do you have for COSP- and for satellite data to determine a daily mean value? Can the averaging procedures produce a bias in the results, maybe the difference in daily averaged cloudiness in Figure 3?*
*What is the physical interpretation of a 'daily mean cloud cover'? 12h cloud free plus 12 h full cloud cover results in 50% cloudiness?*
ANS: subsubsection 2.1.1. is revised to 2.2.
More information regarding the satellites are included in the manuscript. p5, l.9 has been rephrased: However, a meaningful evaluation of modeling with satellite observations is challenging because of the difference in the model variables and the satellite retrievals.
To produce similar output to satellite data, COSP requires grid mean vertical profile of temperature, humidity, hydrometer mixing ratio, cloud optical thickness and emissivity, surface temperature and emissivity from the model. It produces the output comparable

with satellite data in three steps. First it addresses the mismatch between model and satellite pixel, second vertical profiles of individual sub-columns are passed to each instruments and finally COSP statistic module gathers the output from all instruments (Bodas-Salcedo et al., 2011). Since COSP is running online with COSMO model, it is able produce output similar to model simulation (in every hour).

An important aspect of COSP satellite simulator is positional errors due to mismatch between meteorological regimes in the observation and models, which is not considered.

• *p.5 Section 3.1: The synoptic situation should be described for the situation on 17 February, the day of the later discussion and evaluation.*
ANS: Revised as suggested.

• *p.6 l. 16. 'Northerly wind'? Fig. 2 shows mostly south-westerly winds over the Atlantic.*
ANS: Revised.

• *p.6 l. 18-19. Please precise the sentence 'The cold continental air mass ...'.*
ANS: Revised.

• *In Section 3, you use 3 version of the COSMO model and several satellite data sets for mutual intercomparison. Please make clear everywhere, which respective data sets are compared, and break the passages of different intercomparisons. Please use always the same expressions. E.g. p.7 l. 2. Which two model versions? What is the 'MODIS simulator' (also l. 30)?*
ANS: P7, l.2: this line is removed from the manuscript. P7, l.30: We have also included cloud droplet number concentration $N_d$ as a diagnostics of the model via COSP

satellite simulator (MODIS simulator in COSP).

• *Section 3.2 (in particular) contains inconsistencies in wording and notation com-pared to the rest. E.g., optical depth $\delta$ vs. $\tau_c$, COSP satellite simulator vs. MODIS satellite simulation? Please unify.*
ANS: Revised

• *p.7 l.11pp. The spatial structure of the fields are similar. On the linear scale, I would not agree to 'slightly larger' (l. 11) or 'slight underestimation' (l. 18). I am well aware that both data sets are subject to many sources of error, hence a similar field structure and a similar order of magnitude should be acceptable, but not whitewashed.*
*P.7, l. 14pp. The strongest differences do not occur near the Atlantic coast, but in the most western part of the domain. I have the impression that the model does not catch these clouds. Please clarify.*
ANS: P7, l.11, rephrased to: In satellite, it varies between 5 to 54 and in model between 5 to 45, with maximum values observed over similar geographical regions. However, the satellite derived cloud optical depth and liquid water path are overestimated while comparing with model (COSMO-MUSCAT) outputs.
P7, l.14: this sentance has benn removed.

• *p. 7 l. 19. Correct unit of cloud water path.*
ANS: Revised as suggested.

• *p.7 l.20pp Fig.4 g-i. I do not follow your interpretation. The differences should be seen in relation to the signal. The least (relative) difference should be seen in the LWP, since the amount of condensate is primarily determined by other than microphysical processes and is to be seen in relation to the change in cloud ice. The sequel of e.g.,*

*red and blue bands over the Biscaya may be a phase shift. A decrease of re by $10\mu m$ is of the order of the signal, not a 'slight reduction'.*
*Please precise. I agree with your conclusion of l. 27-28. However, I cannot see the superiority of COSMO-MUSCAT from the presente material.*
ANS: Revised

• *p.7 l. 25. Again: Not 'slight' and 'little'.*
ANS: Revised

• *p.8 l.3. 'cloud microphysics are modified'. If this is worth mentioning, then please be more precise*
ANS: Revised

• *p.8 l.3. Please explain what you mean by 'better agreement'. Allgemeine FRAGE!!*
ANS: This part is revised: Even though, the satellite derived $N_d$ has poor spatial distribution, the $N_d$ values are underestimated while comparing with COSMO2M and it is overestimated while comparing with COSMO-MUSCAT

• *p.8 l.6. Fixed CCN = 300 $cm^{-3}$ in COSMO-2M? This is in contradiction to Section 2.1, telling $N_{ccn}$ is given as function of S.*
*l. 32. Similar: 'constant cloud condensation nuclei profile'?? Please clarify.*
ANS: In COSMO-2M $N_{ccn}$ is a function of S, whereas $C_{ccn}$ kept constant. In the coupled model constant $C_{ccn}$ in the two-moment scheme is replaced by gridded $C_{ccn}$ proxy from MUSCAT, which is four dimensional.

• *p.8 l. 9pp. The aerosol NUMBER (not 'mass') concentration is given in Fig. 5c. Could you please comment on the fact, that Sulfate is so much larger than Nd for COSMOMUSCAT? Is the result of Boucher and Lohmann (1995) transferrable to your model concept?*
ANS: The main objective of this paper is the replace the constant $C_{ccn}$, in

COSMO2M(COSMO two-moment), with interactive aerosol from MUSCAT model. Since the MUSCAT model is available with aerosol mass concentration (in this case sulfate aerosols), we have used Boucher and Lohmann parameterization to calculate $C_{ccn}$ number concentration from mass concentration.

- *p.8 l.14pp Please revise the para.*
*'the model exhibits more clear grid points.' What do you mean?*
*'The model is unable to capture sub grid scale cloud patterns': A subgrid scale cloud cannot be captured by the microphysics parameterization of Seifert and Beheng (2012) or similar ones. You would need a different tool.*
*'the satellite may overestimate the retrievals.' What do you mean?*
ANS: This paragraph has been revised. clear grid means, cloud free region, which is also revised in the manuscript.

- *p. 8, Section 3.3. l. 25. '(20 to 20 $Wm^{-2}$' ?? 'some regions': Please precise*
*Fig. 6. The colorbars are differently scaled for most of the subfigures. Sometimes this is straightforeward (e.g., a and f vs. b and f), sometimes, however, confusing (e.g. a vs.c, j vs. l). Please unify the scaling.*
*Please also consider to plot the net UPWARD LWF to have the colors consistent to the SWFs, e.g., blue for weak differences. Same for Fig. 7.*
*Fig. 7 a-d contains is repetition of Fig. 6 e-h. Use the difference fields COSMO2M rad minus CERES instead.*
*l. 27/28. I cannot follow the statement 'the differences are neither systematic nor large'. Please interpret the radiative flux differences also in terms of the cloud properties.*
ANS: Revised to Northern part of the domain. Color scale of Figure 6 & 7 are revised. Line 27/28 has been removed from the manuscript.

- *p. 9 l. 7pp*

*Please check the conclusions with regard to the above points for more precise statements. Conclusion 1. If you refer to the model runs COSMO-2M and COSMO-MUSCAT, please say so. Then, this statement does not agree with p.7 l. 20-29. Please clarify.*
*Conclusion 2. Precise the 'modified model simulation'.*
ANS: Revised as suggested.

• *p9 l21. Missing reference.*
ANS: Revised as suggested.

• *If a paper is written by two authors, please cite as 'A and B (1999)'*
ANS: Revised as suggested.

• *p.11: Citation of IPCC is incomplete.*
ANS: Revised as suggested.

• *Please check ALL figures w.r.t. wording within the plots and in the legends. E.g., in Fig. 2 'Temeperature', in Fig.3 'MUCAT', in Fig. 4 g-i 'CSOMO2M'.*
ANS: Revised as suggested.

[Figure]

**Fig. 1.** COSMO-MUSCAT cloud ice optical properties on 17 February 2007.

Please also note the supplement to this comment:
http://www.geosci-model-dev-discuss.net/gmd-2016-186/gmd-2016-186-AC4-
supplement.pdf

[Figure]

**Fig. 2.**

---

## Referee Report (RR1)

Review of revised version of: *"Implementation of aerosol-cloud interactions in the regional atmosphere-aerosol model COSMO-MUSCAT(5.0) and evaluation using satellite data"*

The clarity and quantitative discussion of the results has improved significantly in the revised paper. In particular the addition of Fig. 5 has helped greatly in the discussion. Furthermore it further justifies focusing the detailed comparison on a single day of the simulated period. I can therefore now recommend the paper to be published at GMD following only minor technical/editorial comments.

Technical Comments:

1) P6L6: COSMO2M, COSMO2MR and COSMO-MUSCAT used here, but COSMO2MR has not yet been introduced. All acronyms are introduced again (in case of COSMO-MUSCAT and COSMO2M) on P6L18ff. Please adjust as necessary.

2) P9L9: Please remove or reword first sentence. The aerosol-cloud-radiation coupling is already discussed in previous sections in context of cloud optical thickness.

3) Fig. 5: Please relabel "overall simulations". Suggestions: either "15-25 Feb 2007", or "10 day period".

4) Fig7: I would suggest to reword the caption to something like: "Net shortwave and longwave radiative flux at the surface and top of atmosphere for COSMO2M (a-d) and COSMO2M (e-h). Differences in radiative fluxes between the two simulations are shown in panels (i-l).

---

## Referee Report (RR2)

*Review of revised version of* **Implementation of aerosol-cloud interactions ......**
*by S. Dipu Sudhakar et al.*

The authors present the revised version of a numerical study on aerosol, clouds and radiation with mutual interactions and compare the results with saltellite derived data.
Although, the paper shows some improvement compared to the 1st version, it is still far from publication. Still, the presentation is partly vague, requires physical interpretation, lacks some corrections according to the suggestions in previous review, and requires a substantial improvement of the language. It is strongly recommended to follow all comments. Slanted fonts stand for citations from the 1st review.
Yet, before publication, I suggest another considerable revision.

**Comments**

- The following 2 questions have not been answered:
  *In midlatitude winter I expect that the ice phase plays an important role in the development of clouds and precipitation (Bergeron-Findeisen effect!), and you use the Seifert and Beheng (2006) scheme for mixed phase clouds. The paper, however, is devoted to the liquid phase alone.*
  *Please discuss the effect of the modified treatment of drop nucleation on the ice phase properties, since a modification in one path of condensate formation is connected with an opposing trend in other path(s).*
  *How do you determine the effective radius under cloud free conditions?*
  Do you use the scheme of Seifert and Beheng (2006) in its warm cloud version? If Yes, then please argue for the neglect of the ice phase in midlatitude winter. If No, then please explain the changes in the ice phase properties in the whole model domain when changing the nucleation treatment. See p.6 l15 'screened for liquid phase clouds only'.
  Next, concerning $r_e$. Do you assume a lower limit of $r_e$ for cloud free conditions (see question above), as suggested in your response? If Yes, $r_{e,min} = 2\mu m$ (see p.6 l15), this - together with $\tau = 5$ - results in $N_d = 5.4 \times 10^9 m^{-3}$ (Eq. 9). Something goes wrong here.......
  Please clarify.

- Equations (3) - (5) as in the first review:
  The reviewer is familiar with the calculation of the moments and related properties from the cloud drop size distribution $\phi(D)$. Then, (4) follows as

$$\lambda = \left[ \frac{\pi \rho_w N \Gamma(\mu + 4)}{6 \rho q_c \Gamma(\mu + 1)} \right]^{1/3}$$

  with $\rho$ air density, $\rho_w$ bulk density of liquid water, $q_c$ mass fraction of liquid water, $N$ number of drops per volume, as was already explained in the 1st review.
  Please revise and check the relationship $\lambda(N, q_c)$ used throughout the paper.

- The following questions have not been answered:
  *Problem of avaraging.*
  *p. 7, Figs. 4, 6(new). Cloud water path is a property defined for the whole air column. Cloud effective radius, cloud droplet number concentration, and sulfate aerosol number concentration are defined locally, and for a grid point model the data are interpreted to be representative for the grid cell. For which level are the presented data relevant? If they are vertical averages, please discuss, how the vertical average is calculated, how cloud free layers are considered, how the result is to be interpreted, etc. This point is*

*even more complicated for the local variable $r_e$, which depends nonlinearly on the local variables $N$ and $q_c$. Likewise, optical thickness is defined for a certain layer of thickness $dz$, maybe the layer where the respective $r_e$ holds. The presented fields (do they hold for the whole column?) depend on the averaging method.*

*The same question arises for the daily averaging procedure and concerns also liquid water path. It concerns both, model and satellite data.*

*Please explain, and correct the discussion where neccessary.*

The reviewer is aware that you use the COSP satellite simulator. The question of averaging, however, is not answered, and the added text p. 5 l.27pp is not helpful in this context.

- As before: interpretation of Figs. 4, (new) 6.
  Drop number concentration, liquid water content and path, optical thickness, and effective radius are interrelated, not independent of each other. The correlation may be positive or negative, see e.g., (5) states $\tau \propto LWP$ and $\tau \propto 1/r_e$, while Fig.4 suggests on first glance only the first relation. Please interprete the graphics in terms of these interrelations.
  Again: For the discussion of the improvement of COSMO-MUSCAT to COSMO-2M it would be helpful to include the COSMO-2M-fields in Fig. 4 besides (or instead of) the difference fields.

- The following questions have not been answered:
  *The choice of the parameters $C_{ccn}$ (p. 4 bottom) is a good general guess, however, not a universal constant. Did you do a similar run with modified $C_{ccn}$-values to check its influence - in opposition to the influence of the full interactive treatment with MUSCAT? COSMO-MUSCAT seems to result in much smoother distributions than COSMO-2M, in particular Fig. 5 (new: Fig. 6). Do you have an explanation?*

- Please go through the whole paper carefully. Avoid repetitions, strengthen the physical interpretation, improve the verbal presentation, eliminate errors in grammer and spelling.

- p.7 l.9: wrong unit of rainfall amount.

- p.7 l.22, Fig. 4: You give the range of data for cloud optical depth with a minimum of 5 for both satellite and model data. This does not agree with the figures: Huge white areas occur, and white stand for data less than 5 according to legend. If on the other hand, you prescribe $\tau = 5$ as minimum (p.6 l.15), than refer to that chosen threshold.
  A similar problem concerns the effective radius (l. 26) with white standing for $r_e < 2$ $\mu$m, that is the mentioned minimum value.

- p.8 l.7pp. Please clarify: The under-/overestimation refers to the frequency of the respective value of liquid water path or optical depth.
  Does the model really overestimate 'low cloud' or does it overestimate the frequency of low optical depth cases?
  Please clarify for all 3 properties.

- I appreciate the inclusion of fig. 5 for the different probability distribution functions. How do you define here the PDF? Unit?
  The interpretation is given as a description of higher/lower PDF and the conclusion that the PDFs are similar for the single day and the period. Unfortunately, an interpretation of the differences/coincidences in the structure of the PDFs for the model and the satellite data is missing. It would be interesting to look for reasons of the shift in PDF for liquid water path, the more frequent occurrence of low $\tau$ in COSMO, and the preponderance

of $r_e$ around 10 $\mu$m in COSMO, the peak in the MODIS-PDF for cloud optical depth between 150 and 200, the drop of COSMO-PDFs to 0 around LWP = 20m $\tau$ = 50, $r_e$ = 30 $\mu$m, and many more features.

What are the PDFs in the COSMO-2M case? This may help the interpretation.

- p.8 l.19p. I do not see this.

  p.8 l.20p: Something goes wrong with the sentence.

  Please clarify.

- p.7 l.20pp (new: p.8 12pp), Fig.4 g-i. You describe what is seen in the figures, but you do not give a physical interpretation. As suggested in the 1st review, the differences should be seen in relation to the signal, and then you find differences of 50% of the signal. If you use the full Seifert and Beheng schene, then the difference in LWP should be seen also in relation to the change in cloud ice concentrations (not only locally but in the whole domain). The sequel of e.g., red and blue bands over the Biscaya may be a phase shift.

  p.8 l.19p. I cannot see the superiority of COSMO-MUSCAT from the presented material.

  p.8 l.20p. Sentence unclear. (i) Any explanation is missing. (ii) If you compare two models differring in 2 parameterizations, you cannot trace back the differences simply to the microphysics parameterization. Please clarify.

- p.8 l.29-31. Under-/Overestimation - in comparison to the what?

  l. 31pp. '... explained by cloud microphysics modification.... MUSCAT-model.' I cannot see an explanation why $N_d$ should be reduced in the MUSCAT-model. Please explain.

- See 1st review. Old: *p.8 l.6, new: l.32 Fixed CCN= 300cm$^{-3}$ in COSMO-2M? This is in contradiction to Section 2.1, telling $N_{ccn}$ is given as function of $S$.*

  *l. 32. Similar: 'constant cloud condensation nuclei profile'??*

  *Please clarify.*

  You have answered the question in your response, but you did not clarify anything in the paper. Please precise your wording and distinguigh clearly between the properties $N_{ccn}$ and $C_{ccn}$ as well as to talking about CCN (an abbreviation to shorten the text).

- As 1st review:

  *p.8 l. 9pp, new: p.8 l.34pp. The aerosol NUMBER (not 'mass') concentration is given in Fig. 6c. Could you please comment on the fact, that $N_{sulfate}$ is so much larger than $N_d$ for COSMO-MUSCAT? Is the result of Boucher and Lohmann (1995) transferrable to your model concept?*

  Please make clear which parameter you are talking about and give a precise explanation.

- As 1st review: *p.8 l.14pp, new: p.9 l6pp. Please revise the para.*

  *'The model is unable to capture sub grid scale cloud patterns': A subgrid scale cloud cannot be captured by the microphysics parameterization of Seifert and Beheng (2006) or similar ones. You would need a different parameterization tool. '*

  You talk of the 'coarse resolution' of the model. I would not call a mesh size of 28 km 'coarse'. More important: Please tell the resolution of the satellite data when you compare the resolution.

  If the snow cover can be ignored - how are the satellite retrievals affected by the snow cover?

  Please use a more precise wording.

- As 1st review: *p. 8, new: p.9, Section 3.3. l. 25 new: l. 19. '(20 to 20 W m$^{-2}$)'??*

  Fig. 7. The colorbars are differently scaled. Sometimes this is straightforeward, but

sometimes, however, confusing. Please unify the scaling insofar as to use the same scaling at least for SFC and TOA net down SWR. Same for Fig. 7.

Fig. 7 a-d contains is repetition of Fig. 6 e-h. Use the difference fields COSMO2M rad minus CERES instead.

Are figs. 7 i and k really different?

Once again: *Please interpret the radiative flux differences also in terms of the cloud properties.*

- p. 10
  Now you have a contradiction in conclusions 1 and 2. Conclusion 1 - 'modification has only a minor effect'. Conclusion 2 - COSMO MUSCAT shows an improvement in the cloud microphysical properties.

  Please clarify.

  What is the outcome of the PDF analysis?

  From the 1st review for Conclusion 1: *If you refer to the model runs COSMO-2M and COSMO-MUSCAT, please say so. Then, this statement does not agree with p.7 l. 20-29 (new: p.8 l.12-21). Please clarify.*

  Conclusion 3. You can find differences in the model runs with and without the effect on radiation. How do you know that the new approach gives results closer to reality?

---

## Author Response (AR2)

**Reviewer # 4**

**Review of revised version of: "Implementation of aerosol-cloud interactions in the regional atmosphere-aerosol model COSMO-MUSCAT(5.0) and evaluation using satellite data"**

The clarity and quantitative discussion of the results has improved significantly in the revised paper. In particular the addition of Fig. 5 has helped greatly in the discussion. Furthermore it further justifies focusing the detailed comparison on a single day of the simulated period. I can therefore now recommend the paper to be published at GMD following only minor technical/editorial comments.

**Technical Comments:**

1) P6L6: COSMO2M, COSMO2MR and COSMO-MUSCAT used here, but COSMO2MR has not yet been introduced. All acronyms are introduced again (in case of COSMO-MUSCAT and COSMO2M) on P6L18ff. Please adjust as necessary.
**ANS:** All acronyms are revised in the manuscript.

2) P9L9: Please remove or reword first sentence. The aerosol-cloud-radiation coupling is already discussed in previous sections in context of cloud optical thickness.
**ANS:** P9L9: Sentence is removed form the manuscript.

3) Fig. 5: Please relabel "overall simulations". Suggestions: either "15-24 Feb 2007", or "10 day period".
**ANS:** Overall simulation has been revised to 15-24 Feb 2007.

4) Fig7: I would suggest to reword the caption to something like: "Net shortwave and longwave radiative flux at the surface and top of atmosphere for COSMO2M (a-d) and COSMO2M (e-h). Differences in radiative fluxes between the two simulations are shown in panels (i-l).
**ANS:** Revised as suggested.

**Reviewer # 5**

**Reviewer Comments:**

Title: Implementation of aerosol-cloud interaction in the regional atmosphere-aerosol model COSMO-MUSCAT and evaluation using satellite data Authors: S. Dipu, J. Quaas, R. Wolke, J. Stoll, A. Muhlbauer, M. Salzmann, B. Heinold, I. Tegen

The manuscript has substantially improved with respect to the first version. That not all changes (additions) to the text are marked in bold face makes the evaluation of the revised manuscript unnecessarily difficult. Some replies were given only as 'reply to reviewer' but should have entered the manuscript as well. Just one example to illustrate the point: while the 'reply to reviewer' now states that Figures 7 and 8 show a 24h average for February 17, this information is not given anywhere in the manuscript.

Evidence for the claimed superiority of COSMO-MUSCAT over COSMO2M is, however, still more on the qualitative than on the quantitative side. With comparatively little effort this could be further improved, I think, and I would encourage the authors to do so. What must be improved in any case is the language. In a number of places in the current manuscript it is not even clear what the authors want to say. Also, there is still an overall lack of precision (see minor pointsbelow).

The manuscript still requires major revisions to meet GMD standards.

**Major points:**
1) The language remains a major issue that must be improved. In several places, it is not even clear what the authors want to say. I give only two examples.

**ANS:** We went through the entire document and clarified the formulations.

p.9, l.8: "Further, the satellite retrieval (mainly thin clouds) are affected by snow cover, which could be rather ignored."

**ANS:** p9,l.8 has been moved to section: "model evaluation method" and rephrased as,
Since the analysis is carried out for winter, satellite retrievals can be affected by snow cover on the ground. However, the MODIS retrieval (Platnick et al., 2001) uses a combination of absorbing spectral channels for which the snow/ice albedo is relatively small which makes it suitable for retrieving cloud properties over snow.

p.9, l.20: "It is also noted that the cloud microphysics radiation coupling results in reduction in cloud optical properties, which would results more downward shortwave and upward longwave especially at the surface."

There are many more sentences that are equally unclear, but I do not consider it my task as a reviewer to list them all. If none of the authors has sufficient command of the English language they should seek assistance from a native speaker.

**ANS:** We carefully copyedited the text in this re-revision, and will also have the publisher help with it where necessary.

2) While improvements were made, the manuscript still lacks precision and relies more on qualitative than quantitative statements. For example, in the abstract it is said that the cloud effective radius shows an increasees of 1 to 4 micro meter and the cloud droplet number concentration is reduced by 100 to 200 cm-3. Where do we see that in the manuscript? I greatly appreciate the newly added pdfs (Figure 5). But if you want to make the point that COSMO-MUSCAT is closer to MODIS than COSMO2M, why not show also COSMO2M in that figure?

**ANS1:** The above sentence in the abstract has been modified to: The cloud effective radius shows an increase of 9.5%, and the cloud droplet number concentration reduced by 21.5%.

**ANS2:** COSMO2M is also included in Figure 5 and discussed in the text.

Another example concerns Figures 7 and 8, comparison of shortwave and longwave fluxes among the different models and with CERES. From looking at the figures you conclude that differences are neither large nor systematic. Equally based on just looking at the figures I would argue that panel 8e (CERES) is most similar to panel 7A (COSMO2M), and differs more strongly from panels 7e (COSMO2Mrad) and 8a (COSMO-MUSCAT). Why not remap all the data on the same grid and provide quantitative estimates (means, pattern correlations, etc) to decide the issue?

**ANS:** Figure 7 and 8 are modified, in the modified manuscript, net shortwave flux at the surface and net longwave flux at the TOP are compared, with different color bars. Also the fluxes are considered for 17 February 2007 and are daily averaged (0hrs to 23.00hrs). Figure R1 shows the spatial correlation between modeled and CERES flues.

**Minor points:**
p.4, l.11: What is the numerical value of mu?
**ANS:** Here, the value of $\mu$ is 2, which is now added to the manuscript as well.

p.5, l.9: What is TNO?
**ANS:** TNO is European emissions processing, and stands for Nederlandse Organisatie voor toegepast-natuurwetenschappelijk onderzoek (Netherlands Organisation for Applied Scientific Research). This is now added to the text.

p.6, l.3: What do you mean by a positional error due to mismatch between meteorological regimes?
**ANS:** We mean the error in location, rather than characteristics of an event in the forecast compared to reality.
p.6, l.25: You may want to add / state explicitly that COSMO-MUSCAT treats radiation in the same way as COSMO-2MR.
**ANS:** Revised as suggested.

p.7, l.9: Rainfall of 100 kg m$^3$? Please clarify.
**ANS:** The unit of rainfall has been corrected to 100mm, which accumulated precipitation over 96 hrs.

p.7, l.22: You write that the cloud optical depth of the satellite data varies between 5 and 54. However, the pdf in Figure 5 shows values larger than 100. Please clarify.
**ANS:** The range of satellite data has been corrected to 5 to 100, althoug most of the values

are lies between 5 to 50.

p.7, l.24: You say that the satellite derived cloud optical depth and liquid water path are overestimated. Do you mean that the satellite overestimates these quantities systematically, for example with respect to surface based observations or reanalysis? If so, please give a reference.

**ANS:** In the case study, the domain averaged cloud optical depth, effective radius and LWP are 23.34, 11.30$\mu$m, 0.175 kg/m$^{-2}$, whereas the COSMO-MUSCAT derived values are 7.60, 9.93$\mu$m, and 0.056 kg/m$^2$, which illustrates the satellite derived cloud optical properties are overestimated (Page 8, l.20-22.).

p.7, l.26: 'cloud droplet radius between 2 and 20 micro meter'. But the pdf in Figure 5 shows values up to 30 micro meter. Please clarify.

**ANS:** COSMO-MUSCAT derived cloud droplet radius varies between 3 and 16 $\mu$m and MODIS effective radius ranges between 2 to 30 $\mu$m, which is same as in Figure 5.

p.8, l.8: 'For cloud optical depth, the model overestimate low clouds (optical depth below 10)...' Do you mean optically thin clouds?

**ANS:** This sentence modified as: The cloud optical depth PDF shows that thin clouds (cloud optical depth $< 10$) in all model versions occur substantially more frequently than in the satellite retrievals, and thick clouds (cloud optical depth $> 30$), less frequently.

p.8, l.12: Why not also show COSMO2M in Figure 5?

**ANS:** COSMO2M is also included in Figure 5

p.8, l.29: '...satellite derive Nd values are overestimated...' Can you give a reference?

**ANS:** Zhang et al., 2012, Storelvmo et al., 2009.

p.9, l.5: 'However, model derived cloud optical properties are well correlate.' With what do they correlate?

**ANS:** Here, the model derived optical properties strongly correlated to MODIS level-2 products, despite the low magnitudes. This is now discussed in the section 3.2 of the manuscript.

Section 3.3: Looking at Figures 7 and 8 the agreement between CERES and the different models for the surface shortwave radiation seems better in the absence of the revised radiation scheme. Please comment.

**ANS:** Figure 7 and 8 are revised in the manuscript with net shortwave flux at the surface and net longwave flux at TOA, and are compared with CERES data.

p.10, l.14: "The satellite retrievals suggest the revised model version is more realistic in both quantities." I find it difficult to draw this conclusion based on Figure 4 (showing cloud effective radius, but only for MODIS and COSMO-MUSCAT) and Figure 6 (showing cloud droplet number, COSMO2M looking more similar (more red) to MODIS than COSMO-MUSCAT). Could you further corroborate your conclusion?

**ANS:** In the revised manuscript, MODIS cloud optical properties are under gone revised cloud screening based on the recommendations by Nakajima and King (1990) (cloud optical depth less than 5 and effective radius less than 2$\mu$m are not considered), in which cloud optical depth, effective radius and LWP are more realistic with satellite observations. Based on the new analysis, the conclusion is also revised.

Conclusions: I think it would be worthwhile to state again that you consider only warm clouds.
**ANS:** Revised as suggested

Figures 2 and 6: Does the 100% cloud cover from Figure 2 together with the close to zero cloud droplet number concentration in Figure 6 imply that the majority of clouds is ice (i.e. not warm / liquid clouds)?
**ANS:** Figure 2 is daily averaged product, where as Figure 6 is averaged for 8:14hrs of the same day. Additionally, $N_d$ is in cloud which is independent of clod fraction and the re-gridding can results in more cloudy regions. Figure R2 shows comparison between ISCCP simulator derived cloud fraction and MODIS simulator derived cloud droplet number concentration.

**References:**

Nakajima, T. and M.D. King, (1990), Determination of the Optical Thickness and Effective Particle Radius of Clouds from Reflected Solar Radiation Measurements. Part I: Theory. J. Atmos. Sci., 47, 1878–1893, doi: 10.1175/1520-0469(1990)047¡1878:DOTOTA¿2.0.CO;2.

Zhang, Y., P. Karamchandani, T. Glotfelty, D. G. Streets, G. Grell, A. Nenes, F. Yu, and R. Bennartz (2012), Development and initial application of the global-through-urban weather research and forecasting model with chemistry (GU-WRF/Chem), J. Geophys. Res., 117, D20206, doi:10.1029/2012JD017966.

Storelvmo, T., U. Lohmann, and R. Bennartz (2009), What governs the spread in short-wave forcings in the transient IPCC AR4 models? Geophys. Res. Lett., 36, L01806, doi:10.1029/2008GL036069.

[Figure]

**Figure R1:** Comparison between short wave and long wave fluxes at surface and top of the atmosphere with CERES satellite fluxes and correlation between satellite (CERES) and models (COSMO2MR and COSMO2M)

[Figure]

**Figure R2:** (a) Model derived cloud fraction (via ISCCP cloud fraction) and (b) cloud optical depth (via MODIS satellite simulator).

**Reviewer # 6**

*Review of revised version of* **Implementation of aerosol-cloud interactions ......** *by S. Dipu Sudhakar et al.*

The authors present the revised version of a numerical study on aerosol, clouds and radiation with mutual interactions and compare the results with saltellite derived data.

Although, the paper shows some improvement compared to the 1st version, it is still far from publication. Still, the presentation is partly vague, requires physical interpretation, lacks some corrections according to the suggestions in previous review, and requires a substantial improvement of the language. It is strongly recommended to follow all comments. Slanted fonts stand for citations from the 1st review.

Yet, before publication, I suggest another considerable revision.

**Comments**
• The following 2 questions have not been answered:

In midlatitude winter I expect that the ice phase plays an important role in the development of clouds and precipitation (Bergeron-Findeisen effect!), and you use the Seifert and Beheng (2006) scheme for mixed phase clouds. The paper, however, is devoted to the liquid phase alone.

Please discuss the effect of the modified treatment of drop nucleation on the ice phase properties, since a modification in one path of condensate formation is connected with an opposing trend in other path(s).

How do you determine the effective radius under cloud free conditions?

Do you use the scheme of Seifert and Beheng (2006) in its warm cloud version? If Yes, then please argue for the neglect of the ice phase in midlatitude winter. If No, then please explain the changes in the ice phase properties in the whole model domain when changing the nucleation treatment. See p.6 l15 'screened for liquid phase clouds only'. Next, concerning re. Do you assume a lower limit of re for cloud free conditions (see question above), as suggested in your response? If Yes, $r_{e,min} = 2\mu m$ (see p.6 l15), this - together with $\tau = 5$ - results in $N_d = 5.4 \times 10^9$ m$^{-3}$ (Eq. 9). Something goes wrong here.......

Please clarify.

**ANS:** In our models versions we have used the Seifert and Beheng (2006) scheme for mixed phase clouds, in which the ice-phase are also affected by the new droplet activation parameterisation, e.g. due to Bergeron-Findeisen process. However, we have used the Seifert and Beheng (2006) in warm cloud only. In the Seifert and Beheng scheme, the droplet nucleation scheme is parameterized as,

$$\left.\frac{\partial N_c}{\partial t}\right|_{nuc} = \begin{cases} C_{ccn}kS^{k-1}\frac{\partial S}{\partial z}w, & \text{if } S \geq 0, w\frac{\partial S}{\partial z} > 0, \\ & \text{and } S < S_{max}, \\ 0 & \text{else.} \end{cases} \tag{1}$$

and the ice nucleation rate is given by

$$\left.\frac{\partial N_i}{\partial t}\right|_{nuc} = \begin{cases} \frac{N_{IN}(S_i,T)-N_i}{\Delta t}, & \text{if } S_i \geq 0 \text{ and } N_i < N_{IN}(S_1,T) \\ 0 & \text{else.} \end{cases} \tag{2}$$

The modification is done only with the $C_{ccn}$ of the equation 2, whereas $N_{IN}$ is not coupled with prognostic aerosols. Even though, the $C_{ccn}$ modification would modify the ice phase, the effect is very less compared to the liquid phase of the cloud. Figure R1, shows the ice cloud optical

properties for different model versions.

In the revised analysis, the effective radius are not diagnosed over cloud free regions.

Since the modification have done for the liquid phase of the clouds, for comparison, satellite cloud products are screened for the liquid phase clouds only.

Again, the screening of the satellite products has been corrected in the revised manuscript, in which the cloud optical properties are considered only if the cloud optical depth $> 5$ and effective radius $> 2$ $\mu$m for a given pixel. This would eliminate high $N_d$ values.

• Equations (3) - (5) as in the first review:
The reviewer is familiar with the calculation of the moments and related properties from the cloud drop size distribution $\phi(D)$. Then, (4) follows as,

$$\lambda = \left[ \frac{\pi \rho_w N \Gamma (\mu + 4)}{6 \rho q_c \Gamma (\mu + 1)} \right]^{\frac{1}{3}} \tag{3}$$

with $\rho$ air density, $\rho_w$ bulk density of liquid water, $q_c$ mass fraction of the liquid water, N number of drops per volume, as was already explained in the 1st review.
Please revise and check the relationship $\lambda(N, qc)$ used throughout the paper.
**ANS:** Equation 4 has been modified in the revised manuscript as

$$\lambda = \left[ \frac{\pi \rho_w N \Gamma (\mu + 4)}{6 \rho q_c \Gamma (\mu + 1)} \right]^{\frac{1}{3}} \tag{4}$$

• The following questions have not been answered:
Problem of avaraging.
7, Figs. 4, 6(new). Cloud water path is a property defined for the whole air column. Cloud effective radius, cloud droplet number concentration, and sulfate aerosol number concentration are defined locally, and for a grid point model the data are interpreted to be representative for the grid cell. For which level are the presented data relevant? If they are vertical averages, please discuss, how the vertical average is calculated, how cloud free layers are considered, how the result is to be interpreted, etc. This point is even more complicated for the local variable re, which depends nonlinearly on the local variables N and qc. Likewise, optical thickness is defined for a certain layer of thickness dz, maybe the layer where the respective re holds. The presented fields (do they hold for the whole column?) depend on the averaging method.
The same question arises for the daily averaging procedure and concerns also liquid water path. It concerns both, model and satellite data.
Please explain, and correct the discussion where necessary.
The reviewer is aware that you use the COSP satellite simulator. The question of averaging, however, is not answered, and the added text p. 5 l.27pp is not helpful in this context.
**ANS:** Cloud optical depth, effective radius and liquid water path are derived using the MODIS simulator which is included in the COSP satellite simulator. Figures 4 and 6 shows the COSP derived MODIS products and its estimations are discussed below, also included in the manuscript.

The MODIS satellite simulator uses profiles of particle size for liquid and ice and corresponding optical depths within each layer of sub-column as a function of model levels. Using the cloud overlap assumption, zero or one cloudiness in each sub-column is created in each level. The

diagnostics are then integrated over the cloudy sub-columns to obtain in-cloud average cloud optical depth and liquid water path. In turn, cloud effective radius is sampled at the cloud tops, which is not a vertical integral. Further, the ISCCP simulator aggregates pixel scale cloud retrievals (fraction of the sub-column with $\tau \geq 0.3$) to estimate cloud fraction (more details: Pincus et al., 2012). Further, cloud droplet number concentration is estimated using equation 9.

To compare with MODIS satellite observation, cloud optical properties (optical depth, effective radius, and droplet number concentration) are averaged for the time 8.00 to 14.00 hrs, which is approximate MODIS Terra overpass time over the domain.

• As before: interpretation of Figs. 4, (new) 6.
Drop number concentration, liquid water content and path, optical thickness, and effective radius are interrelated, not independent of each other. The correlation may be positive or negative, see e.g., (5) states $\tau \propto$ LWP and $\tau \propto 1/r_e$, while Fig.4 suggests on first glance only the first relation. Please interpret the graphics in terms of these interrelations.
Again: For the discussion of the improvement of COSMO-MUSCAT to COSMO-2M it would be helpful to include the COSMO-2M-fields in Fig. 4 besides (or instead of) the difference fields.

**ANS1:** Interpretation of Figure 4 and 6 is given below.

From the above analysis, it can be inferred that COSMO-MUSCAT can be used as a tool for regional aerosol-cloud interaction estimates.The interactive aerosol results show an increase in the cloud droplet effective radius by 9.5% and a reduction in $N_d$ by 21.5%. This indicates that the interactive aerosols in COSMO-MUSCAT model accounts for the increase in cloud droplet size by enhancing the cloud droplet growth and it accounts for lower $N_d$, which attributed to implicit aerosol cloud interactions in the model. Additionally, it reveals the importance of satellite simulators in weather forecast models, which is very efficient for validating model results with different satellite observations. Even though the degree of uncertainty can go together hand in hand with satellite observed and model derived cloud optical properties, the modeled cloud optical properties are in agreement with satellites data.

As for the influence of both LWP and $r_e$ on cloud optical thickness: By construction, the dependencies as the reviewer suggests are relevant in both, observational and model datasets. In the model, we implemented the dependency of cloud optical thickness on $r_e$ (please see Manuscript Section 3.3). In the satellite retrievals, LWP is not retrieved independently, but computed from cloud optical thickness and effective radius. However, the reviewer is of course correct in observing that the LWP variability substantially more impacts the cloud optical thickness variability than the variability in $r_e$.
**ANS2:** Authors think that rather than including COSMO-2M fields, the difference can give a quantitative information about the model modifications. In addition, for reviewers consideration COSMO-2M fields are included in the Figure R2.

• The following questions have not been answered:
The choice of the parameters $C_{ccn}$ (p. 4 bottom) is a good general guess, however, not a universal constant. Did you do a similar run with modified $C_{ccn}$-values to check its influence - in opposition to the influence of the full interactive treatment with MUSCAT? COSMO-MUSCAT seems to result in much smoother distributions than COSMO-2M, in particular Fig. 5 (new: Fig. 6). Do you have an explanation?

**ANS:** Sensitivity experiments with different $C_{ccn}$ values are already reported by Seifert et al.,

2012, which uses low $C_{ccn} = 100$ cm$^{-3}$ and high $C_{ccn} = 3200$ cm$^{-3}$ and it illustrates the influence of different $C_{ccn}$ values. The main difference between COSMO-2M and COSMO-MUSCAT is to replace constant $C_{ccn}$ value with gridded $C_{ccn}$ proxies computed using Boucher and Lohmann parameterization. Hence, the smoother distribution can be explained by activation of more cloud droplets in the COSMO-MUSCAT simulations. Figure 6d is a representative of aerosols in the domain, and it can very with the model levels and time. So the smooth distribution can be due to the temporally and spatially varying aerosols, which results in more droplet nucleation.

• Please go through the whole paper carefully. Avoid repetitions, strengthen the physical interpretation, improve the verbal presentation, eliminate errors in grammer and spelling.
**ANS:** We did go through the entire manuscript with careful copy-editing..

• p.7 l.9: wrong unit of rainfall amount.
**ANS:** Revised as 100mm, which is accumulated precipitation over 96 hrs.

• p.7 l.22, Fig. 4: You give the range of data for cloud optical depth with a minimum of 5 for both satellite and model data. This does not agree with the figures: Huge white areas occur, and white stand for data less than 5 according to legend. If on the other hand, you prescribe $\tau = 5$ as minimum (p.6 l.15), than refer to that chosen threshold. A similar problem concerns the effective radius (l. 26) with white standing for $r_e$ ¡ 2 $\mu$m, that is the mentioned minimum value.
**ANS:** Figure 4 is corrected, since the retrieval error is more for effective radius less than 2 $\mu$m and cloud optical depth less than 5, the satellite retrievals are screened for above threshold values. This part is now corrected in the manuscript.

• p.8 l.7pp. Please clarify: The under-/overestimation refers to the frequency of the respective value of liquid water path or optical depth.
Does the model really overestimate 'low cloud' or does it overestimate the frequency of low optical depth cases?
Please clarify for all 3 properties.
**ANS:** Under/overestimation refers to the frequency of occurrence. This is now explained in the manuscript.

The statistical distribution of satellite and the model cloud microphysical properties are compared and evaluated in terms of probability density functions (PDFs). Figure 5 represents the probability density function of the spatiotemporal distribution of cloud optical depth, effective radius, and liquid water path, defined as the normalized count of occurrence per bin width of cloud optical property. The cloud optical depth PDF shows that thin clouds (cloud optical depth < 10) in all model versions occur substantially more frequently than in the satellite retrievals, and thick clouds (cloud optical depth > 30), less frequently. The modeled cloud effective radius PDFs is constrained to 3 and 16 $\mu$m, where as the satellite retrievals shows a range of 4 to 30 $\mu$m. A shift of the PDF is found in the COSMO-MUSCAT derived PDFs, which indicates the increased droplet size for the interactive Cccn. For liquid water path, modeled PDFs overestimates the clouds with low liquid water path and underestimates clouds with high water paths. The differences in PDFs largely follow what is found for the cloud optical depth, but model deficiencies compared to the satellite retrievals are substantially larger. The analysis also illustrates an increased in cloud optical PDF from COSMO-MUSCAT simulation. Certainly, the drop and preponderance of modeled cloud optical properties can be influenced by model tuning, an approach which, however, hasn't been performed yet for the

COSMO-MUSCAT model version.

• I appreciate the inclusion of fig. 5 for the different probability distribution functions. How do you define here the PDF? Unit?
The interpretation is given as a description of higher/lower PDF and the conclusion that the PDFs are similar for the single day and the period. Unfortunately, an interpretation of the differences/coincidences in the structure of the PDFs for the model and the satellite data is missing. It would be interesting to look for reasons of the shift in PDF for liquid water path, the more frequent occurrence of low $\tau$ in COSMO, and the preponderance of $r_e$ around 10 $\mu$m in COSMO, the peak in the MODIS-PDF for cloud optical depth between 150 and 200, the drop of COSMO-PDFs to 0 around LWP = 20m $\tau = 50$, $r_e = 30$ $\mu$m, and many more features. What are the PDFs in the COSMO-2M case? This may help the interpretation.

**ANS:** These plots represent the density of COD, CER and LWP, which means the normalized count of occurence per bin width of optical property. The density has a unit inverse to that of the bin width.

The shift in the PDF for MODIS cloud optical properties (especially LWP) can arise from the quality filtering of the data, also the log scale is responsible for large shit, which is revised in the manuscript.
The peak in the MODIS cloud optical depth around 150-200 can be due to high retrieval uncertainties towards high cloud optical depth, which is screened in the revised analysis.
the drop and preponderance of modeled cloud optical properties can be can be influenced by model tuning.

• p.8 l.19p. I do not see this.
p.8 l.20p: Something goes wrong with the sentence.
Please clarify.
**ANS:** p.8 l.19p. This sentence is removed from the manuscript
p.8 l.20p: The sentence is modified to : Likewise, the model derived $N_d$ is also estimated using equation 9, which uses COSP (MODIS simulator) derived cloud optical depth and effective radius.

• p.7 l.20pp (new: p.8 12pp), Fig.4 g-i. You describe what is seen in the figures, but you do not give a physical interpretation. As suggested in the 1st review, the differences should be seen in relation to the signal, and then you find differences of 50% of the signal. If you use the full Seifert and Beheng schene, then the difference in LWP should be seen also in relation to the change in cloud ice concentrations (not only locally but in the whole domain). The sequel of e.g., red and blue bands over the Biscaya may be a phase shift.
p.8 l.19p. I cannot see the superiority of COSMO-MUSCAT from the presented material. p.8 l.20p. Sentence unclear. (i) Any explanation is missing. (ii) If you compare two models differring in 2 parameterizations, you cannot trace back the differences simply to the microphysics parameterization. Please clarify.
**ANS:** Physical interpretation: In COSMO-MUSCAT, the aerosol coupling leads to an incrrease in the cloud optical depth by 4.1%, the cloud effective radius by 9.5% and the cloud water path by 14.2%. This implies that gridded aerosols result in increaseing the cloud droplet size, optical depth and LWP. However, this would results in the reduction Cloud droplet number concentrations ($N_d$). This incicates that, as the drolpte size increase, $N_d$ decreses and it would explin the implicit aerosol-cloud interaction in COSMO-MUSCAT.

Please see answer to the first comment above.

Figure R2 shows the MODIS simulator derived cloud optical properties for the COSMO2M Simulations (g-i). From this Figure, it is clear that, there is no phase shift over the Biscaya, However, it can be noticed that, there is an increase in the COSMO-MUSCAT derived cloud optical properties.

Figure R1 (ice optical properties) clearly indicates that, ice phase shows very low signal, which points the fact that, cloud microphysics modification has very little effect on ice cloud optical properties.

p.8 l.19p. I cannot see the superiority of COSMO-MUSCAT from the presented material:
**ANS:** In COSMO-MUSCAT aerosol coupling leads to an increase in the cloud optical depth by 4.1%, the cloud effective radius by 9.5% and the cloud water path by 14.2% which explain the scope of aerosol-could interaction in regional modeling.
p.8 l.20p: The sentence modified to : Likewise, model derived $N_d$ is also estimated using eqn 9, which uses COSP (MODIS simulator) derived cloud optical depth and effective radius.

• p.8 l.29-31. Under-/Overestimation - in comparison to the what?
l. 31pp. '... explained by cloud microphysics modification.... MUSCAT-model.' I cannot see an explanation why $N_d$ should be reduced in the MUSCAT-model. Please explain.
**ANS :** p.8 l.29-31.- On 17 February 2007, the domain averaged CDNC values are 150, 120, and 378 cm$^{-3}$, respectively for COSMO2M, COMSO-MUSCAT and MODIS, which indicates an underestimation of model derived values (Figure 6a-c) as compared to MODIS. Further, from model inter-comparison (COSMO2M and COSMO-MUSCAT), it can be inferred that COMSO-MUSCAT derived CDNC is reduced by 21.5% (Figure 6a and 6b). This may be explained by implicit aerosol cloud interaction. From figure 5, it is also noticed that, there is an increase in cloud droplet size in COSOMO-MUSCAT, this would result in lower CDNC, which attribute to aerosol-cloud interaction in the COMOSO-MUSCAT model.

• See 1st review. Old: p.8 l.6, new: l.32 Fixed CCN= 300 cm$^{-3}$ in COSMO-2M? This is in contradiction to Section 2.1, telling Nccn is given as function of S.
l. 32. Similar: 'constant cloud condensation nuclei profile'??
Please clarify.
You have answered the question in your response, but you did not clarify anything in the paper. Please precise your wording and distinguish clearly between the properties Nccn and Cccn as well as to talking about CCN (an abbreviation to shorten the text).
**ANS:** In COSMO-2M, $N_{cn}$ (activated cloud droplets) is a function of S, whereas $C_{ccn}$ kept constant. In the coupled model constant $C_{ccn}$ is replaced by gridded $C_{ccn}$ proxy from MUSCAT, which is a four dimensional variable.
This is included in the manuscript.

• As 1st review:
p.8 l. 9pp, new: p.8 l.34pp. The aerosol NUMBER (not 'mass') concentration is given in Fig. 6c. Could you please comment on the fact, that Nsulfate is so much larger than Nd for COSMO-MUSCAT? Is the result of Boucher and Lohmann (1995) transferrable to your model concept?
Please make clear which parameter you are talking about and give a precise explanation.
**ANS:** p.8 l.34pp: It is revised to aerosol number concentration.

Figure 6d in the manuscript shows the spatial distribution of sulfate aerosol number concentration (aerosol number concentration proxy) below the convective cloud base (representative of aerosols in the model and it is also averaged for 8-14 hours on 17 February 2007), where high number concentrations are simulated over southeastern Europe. On contrast, $N_d$ are smaller over the same region. This is because the Boucher and Lohmann (1995) parameterization models saturation of $N_d$ over high aerosol or polluted regions (Penner et al., 2001) and the high pressure in this region results in trapping aerosol in the boundary layer.

• As 1st review: p.8 l.14pp, new: p.9 l6pp. Please revise the para.
'The model is unable to capture sub grid scale cloud patterns': A subgrid scale cloud cannot be captured by the microphysics parameterization of Seifert and Beheng (2006) or similar ones. You would need a different parameterization tool. '
You talk of the 'coarse resolution' of the model. I would not call a mesh size of 28 km 'coarse'. More important: Please tell the resolution of the satellite data when you compare the resolution.
If the snow cover can be ignored - how are the satellite retrievals affected by the snow cover? Please use a more precise wording.
**ANS:** In the revised manuscript, this part is revised and the revision is as follows.

From the above analysis, it can be inferred that COSMO-MUSCAT can be used as a tool for regional aerosol-cloud interaction estimates. Because the interactive aerosol coupling results show an increase in cloud droplet effective radius by 9.5% and a reduction in $N_d$ by 21.5%. This indicates that the interactive aerosols in COSMO-MUSCAT model accounts for the increase in cloud droplet size by enhancing the cloud droplet growth and it accounts for lower $N_d$, which attributed to implicit aerosol cloud interactions in the model. Additionally, it reveals the importance of satellite simulators in weather forecast models, which is very efficient for validating model results with different satellite observations. Even though the degree of uncertainty can go together hand in hand with satellite observed and model derived cloud optical properties, the modeled cloud optical properties are in agreement with satellites data.

The effect of snow cover on cloud retrials is included in the model evaluation methods, which is follows,
Since the analysis is carried out for winter, satellite retrievals can be affected by snow cover on the ground. However, the MODIS retrieval (Platnick et al., 2001) uses a combination of absorbing spectral channels for which the snow/ice albedo is relatively small which makes it suitable for retrieving cloud properties over snow.

• As 1st review: p. 8, new: p.9, Section 3.3. l. 25 new: l. 19. Fig. 7. The colorbars are differently scaled. Sometimes this is straightforeward, but sometimes, however, confusing. Please unify the scaling insofar as to use the same scaling at least for SFC and TOA net down SWR. Same for Fig. 7.
Fig. 7 a-d contains is repetition of Fig. 6 e-h. Use the difference fields COSMO2M rad minus CERES instead.
Are figs. 7 i and k really different?
Once again: Please interpret the radiative flux differences also in terms of the cloud properties.
**ANS:** Section 3.3 is revised in the manuscript. In the revised version, we have used net shortwave at the surface and net long wave at the top of the atmosphere (TOA), with different color bars. Also the fluxes are considered for 17 February 2007 and are daily averaged (0hrs to 23.00hrs).
**ANS:** In the revised manuscript, Figure 7 is moved to Figure 9, which shows the comparison

between the COSMO-MUSCAT and CERES derived fluxes (net downward shortwave flux at the surface and net downward longwave flux at the top of the atmosphere (TOA)).

Figure 7i and k are revised in the manuscript, which are not similar.

• p. 10
Now you have a contradiction in conclusions 1 and 2. Conclusion 1 - 'modification has only a minor effect'. Conclusion 2 - COSMO MUSCAT shows an improvement in the cloud microphysical properties.

Please clarify.

What is the outcome of the PDF analysis?

From the 1st review for Conclusion 1: If you refer to the model runs COSMO-2M and COSMO-MUSCAT, please say so. Then, this statement does not agree with p.7 l. 20-29 (new: p.8 l.12-21). Please clarify.

Conclusion 3. You can find differences in the model runs with and without the effect on radiation. How do you know that the new approach gives results closer to reality?

**ANS:** Conclusions have been modified:

A case study has been carried out to compare the model output with observations. The incorporated COSP satellite simulator serves as a link between model and satellite comparisons. Despite the resolution, COSP derived ISCCP cloud fraction shows similar spatial pattern and magnitude. Further, MODIS level-2 cloud optical products such as cloud optical depth, effective radius, and liquid water path are compared. The COSMO-MUSCAT derived cloud optical properties show a similar spatial distribution compared to the MODIS observation. In COSMO-MUSCAT, the cloud optical depth has been increased by 4.1%, cloud droplet effective has been increased by 9.5%, and liquid water path has been increased by 14.2% in comparison to CSOMO2M. In turn, the cloud droplet number concentration estimated from COSMO-MUSCAT model shows a reduction of 21.5% compared to the COSMO2M model. Furthermore, considerable changes in the radiation budget have been found. This analysis indicates that the coupled model (COSMO-MUSCAT) with interactive aerosol treatment results in an increase in cloud droplet size and reduction in cloud droplet number concentration by activation and growth of droplets, which illustrates implicit aerosol-cloud interactions. Also, the cloud properties in COSMO-MUSCAT agree reasonably well with observations, so that it can be used for regional aerosol-cloud interaction studies.

[Figure]

**Figure R1:** COSMO-MUSCAT cloud ice optical properties on 17 February 2007.

[Figure]

**Figure R2:** Cloud optical properties on 17 February 2007, top panel: MODIS level 2 products, bottom panel: COSMO-MUSCAT simulated cloud products, and bootom panel: COSMO2M derived cloud products. All model products are averaged between 08:00 to 14:00 hrs, which is the MODIS aqua overpass time ove the domain.

[revised manuscript text omitted]